# DEEP REPRESENTATIONS FOR TIME-VARYING BRAIN DATASETS

## ABSTRACT

Finding an appropriate representation of dynamic activities in the brain is crucial for many downstream applications. Due to its highly dynamic nature, temporally averaged fMRI (functional magnetic resonance imaging) cannot capture the whole picture of underlying brain activities, and previous works lack the ability to learn and interpret the latent dynamics in brain architectures. In this paper, we build an efficient graph neural network model that incorporates both region-mapped fMRI sequences and structural connectivities obtained from DWI (diffusion-weighted imaging) as inputs. Through novel sample-level adaptive adjacency matrix learning and multi-resolution inner cluster smoothing, we find good representations of the latent brain dynamics. We also attribute inputs with integrated gradients, which enables us to infer (1) highly involved brain connections and subnetworks for each task (2) keyframes of imaging sequences along the temporal axis, and (3) subnetworks that discriminate between individual subjects. This ability to identify critical subnetworks that characterize brain states across heterogeneous tasks and individuals is of great importance to neuroscience research. Extensive experiments and ablation studies demonstrate our proposed method's superiority and efficiency in spatial-temporal graph signal modeling with insightful interpretations of brain dynamics.

## 1 INTRODUCTION

Neuroimaging techniques such as fMRI (functional magnetic resonance imaging) and DWI (diffusion-weighted imaging) provide a window into complex brain processes. Yet, modeling and understanding these signals has always been a challenge. Network neuroscience (Bassett & Sporns, 2017) views the brain as a multiscale networked system and models these signals in their graph representations: nodes represent brain ROIs (regions of interest), and edges represent either structural or functional connections between pairs of regions.

With larger imaging datasets and developments in Graph Neural Networks (Scarselli et al., 2009), recent works leverage variants of the graph deep learning, modeling brain signals with data-driven models and getting rid of Gaussian assumptions typically existed in linear models (Zhang et al., 2019; Li et al., 2019). These methods are making progress on identifying physiological characteristics and brain disorders: In Kim & Ye (2020), authors combine grad-CAM (Selvaraju et al., 2017) and GIN (Xu et al., 2018) to highlight brain regions that are responsible for gender classification with resting-state fMRI data. Li et al. (2020) utilizes the regularized pooling with GNN to identify fMRI biomarkers. Noman et al. (2021) embeds both topological structures and node signals of fMRI networks into low-dimensional latent representations for a better identification of depression. However, the first two works use time-averaged fMRI, losing rich dynamics in the temporal domain. The third combines nodes' temporal and feature dimensions instead of handling them separately, leading to a suboptimal representation (as discussed in section 3.2). To overcome these issues, we propose ReBraiD (Deep **Re**presentations for Time-varying **Brai**n **D**atasets), a graph neural network model that jointly models dynamic functional signals and structural connectivities, leading to a more comprehensive deep representation of brain dynamics.

To simultaneously encode signals along spatial and temporal dimensions, some notable works in traffic prediction and activity recognition domains such as Graph WaveNet (Wu et al., 2019b) alternate TCN (temporal convolution network) (Lea et al., 2016) and GCN (graph convolutional network)

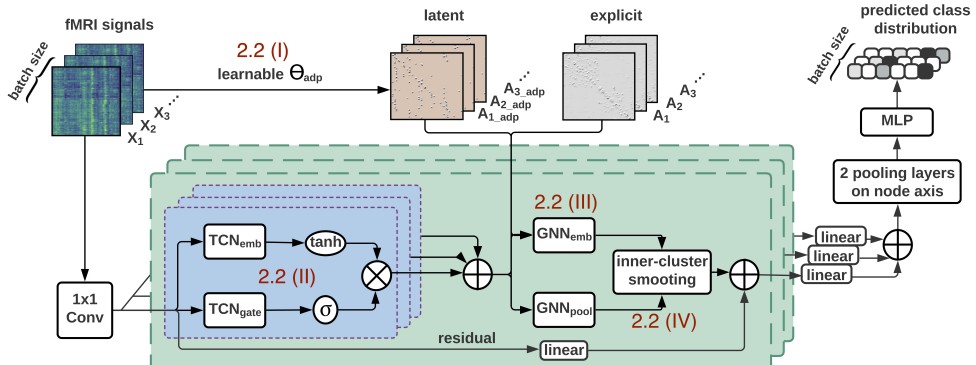

Figure 1: The proposed ReBraiD model for integrating brain structure and dynamics (the architecture shown is for classification). For each batch with batch size $B$, input $X$ has a dimension of $(B, 1, N, T)$[1], and $A, A_{\text{adp}}$ both have the dimension $(B, N, N)$. The encoder (green part) encodes temporal and spatial information alternatively, producing a latent representation in $(B, d_{\text{latent}}, N, 1)$. These embeddings are followed by linear layers for pooling and classification. The final output has a dimension of $(B, C)$.

(Kipf & Welling, 2017). Others (Song et al., 2020; Liu et al., 2020) use localized spatial-temporal graph to embed both domains' information in this extended graph. There are also works incorporating gated recurrent networks for the temporal domain such as (Seo et al., 2018; Ruiz et al., 2020). We choose the first option for ReBraiD, as it is more memory and time efficient, and can support much longer inputs. We also explore the best option when alternating spatial and temporal layers for encoding brain activities with extensive ablation studies. Upon this structure, we propose novel sample-level adaptive adjacency matrix learning and multi-resolution inner cluster smoothing, both of which learn and refine latent dynamic structures. We also make the model more efficient while being effective with the choice of the temporal layer.

Equally important as finding a good representation of brain dynamics is interpreting them. We utilize integrated gradients (Sundararajan et al., 2017) to identify how brain ROIs participate in various processes. This can lead to better behavioral understanding, biomarker discoveries, and characterization of individuals or groups with their brain imagings. We also make the novel contribution of identifying temporally important frames with graph attribution techniques; this can enable more fine-grained temporal analysis around keyframes when combined with other imaging modalities such as EEG (electroencephalogram). In addition, our subject-level and group-level attribution studies unveil heterogeneities among ROIs, tasks, and individuals.

## 2 METHOD

### 2.1 PRELIMINARIES

We utilize two brain imaging modalities mapped onto a same coordinate: SC (structural connectivity) from DWI scans, and time-varying fMRI scans. We represent them as a set of $L$ graphs $\mathcal{G}_i = (A_i, X_i)$ with $i \in [1, L]$, in which $A_i \in \mathbb{R}^{N \times N}$ represents normalized adjacency matrix with an added self-loop: $A_i = \tilde{D}_{\text{SC}_i}^{-\frac{1}{2}} \tilde{\text{SC}}_i \tilde{D}_{\text{SC}_i}^{-\frac{1}{2}}$, $\tilde{\text{SC}}_i = \text{SC}_i + I_N$ and $\tilde{D}_{\text{SC}_i} = \sum_w (\tilde{\text{SC}}_i)_{vw}$ is the diagonal node degree matrix. Graph signal matrix obtained from fMRI scans of the $i^{th}$ sample is represented as $X_i \in \mathbb{R}^{N \times T}$. Here $N$ is the number of nodes, and each node represents a brain region; $T$ is the input signal length on each node. Our objective focuses on classifying brain signals $\mathcal{G}_i$ into one of $C$ task classes through learning latent graph structures.

### 2.2 MODEL

ReBraiD takes $(A, X)$ as inputs, and outputs task class predictions. The overall model structure is shown in fig. 1. For the $i^{th}$ sample $X_i \in \mathbb{R}^{N \times 1 \times T}$, the initial $1 \times 1$ convolution layer increases

---

[1]Axis order follows PyTorch conventions. Dimension at the second index is the expanded feature dimension.

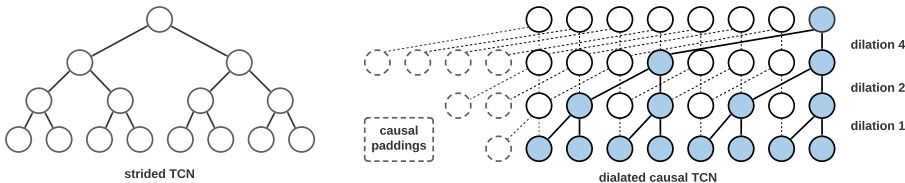

Figure 2: Comparison of strided non-causal TCN (left) and dilated causal TCN (right). The causal aspect is achieved through padding (kernel_size $- 1$) $\times$ dilation zeros to the layer's input. The resulting $\mathbf{y}$ always has the same length as input $\mathbf{x}$, in which $y_\tau$ only depends on inputs $\mathbf{x}_{t \leq \tau}$. We can view strided non-causal TCN as the rightmost node of a dilated causal TCN.

its hidden feature dimension to $d_{h1}$, outputting $(N, d_{h1}, T)$. The encoder then encodes temporal and spatial information alternately, reducing information to feature axis and generating a hidden representation of size $(N, d_{h2}, 1)$. The encoder is followed by two linear layers to perform node embedding pooling and two MLP layers for classification. Cross entropy is used as the loss function: $L_{CE}^{(i)} = -\sum_C y_i \log \hat{y}_i$, where $y_i \in \mathbb{R}^C$ is the one-hot vector of ground truth task label and $\hat{y}_i \in \mathbb{R}^C$ is model's predicted distribution. We now explain different components of the model.

**(I) Learning sample-level latent graph structures.** Structural scans serve as our graph adjacency matrices. However, they are static not only across temporal frames but also across different tasks. In contrast, functional connectivities (FC) are highly dynamic, as shown in appendix A.1.1. To better capture dynamic graph structures, we assign an adaptive adjacency matrix to each input graph signal. Unlike other works such as Wu et al. (2019b) that use a universal latent graph structure, our model do not assume all samples share the same latent graph. Instead, in ReBraiD, each sample has a unique latent structure reflecting its signal status. This implies that the latent adjacency matrix cannot be treated as a learnable parameter as a part of the model. To solve this, we minimize the assumption down to a shared projection $\Theta_{\mathrm{adp}}$ that projects each input sequence into an embedding space and use this embedding to generate the latent graph structure. $\Theta_{\mathrm{adp}}$ can be learned in an end-to-end manner. The generated adaptive adjacency matrix for the $i^{th}$ sample can be written as, where Softmax is applied column-wise:

$$A_{i\_\mathrm{adp}} = \mathrm{Softmax}\left(\mathrm{ReLU}\left((X_i \Theta_{\mathrm{adp}})(X_i \Theta_{\mathrm{adp}})^\top\right)\right), \Theta_{\mathrm{adp}} \in \mathbb{R}^{T \times h_{\mathrm{adp}}} \tag{1}$$

**(II) Gated TCN (Temporal Convolutional Network).** To encode temporal information, we use the gating mechanism as in Oord et al. (2016) in our temporal layers: $H^{(l+1)} = \tanh\left(\mathrm{TCN}_{\mathrm{emb}}(H^{(l)})\right) \odot \sigma\left(\mathrm{TCN}_{\mathrm{gate}}(H^{(l)})\right)$, where $H^{(l)} \in \mathbb{R}^{N \times d \times t}$ is one sample's activation matrix of the $l^{th}$ layer, $\odot$ denotes the Hadamard product, and $\sigma$ is the Sigmoid function. Different from TCNs generally used in sequence to sequence models that consist of dilated $\mathrm{Conv1d}$ and causal padding along the temporal dimension (van den Oord et al. (2016)), we simply apply $\mathrm{Conv1d}$ with kernel 2 and stride 2 as our $\mathrm{TCN}_{\mathrm{emb}}$ and $\mathrm{TCN}_{\mathrm{gate}}$ to embed temporal information. The reason is twofold: first, for a sequence to sequence model with a length-$T$ output, $y_\tau$ should only depend on $x_{t \leq \tau}$ to avoid information leakage, and causal convolution can ensure this. In contrast, our model's task is classification, and the goal of our encoder along the temporal dimension is to embed signal information into the feature axis while reducing temporal dimension to 1. The receptive field of this single temporal point (with multiple feature channels) is meant to be the entire input sequence. Essentially, our TCN is the same as the last output node of a kernel 2 causal TCN whose dilation increases by 2 at each layer (fig. 2). Second, from a practical perspective, directly using strided non-causal TCN works the same as using dilated causal TCNs and taking the last node, while simplifies the model structure and reduces training time to less than a quarter.

**(III) Graph Network layer.** In our model, every $l$ temporal layers (appendix A.2.3 studies the best $l$ to choose) are followed by a spatial layer to encode signals with the graph structure. Building temporal and spatial layers alternatively helps spatial modules to learn embeddings at different temporal scales, and this generates better results than putting spatial layers after all temporal ones.

To encode spatial information, Kipf & Welling (2017) uses first-order approximation of spectral filters to form the layer-wise propagation rule of a GCN layer: $H^{(l+1)} = \mathrm{GCN}(H^{(l)}) = f(AH^{(l)}W^{(l)})$. It can be understood as spatially aggregating information among neighboring nodes to form new node embeddings. In the original setting without temporal signals, $H^{(l)} \in \mathbb{R}^{N \times d}$ is the activation matrix of $l^{th}$ layer, $A \in \mathbb{R}^{N \times N}$ denotes the normalized adjacency matrix with self-connections

as discussed in section 2.1, $W^{(l)} \in \mathbb{R}^{d \times d'}$ is learnable model parameters, and $f$ is a nonlinear activation function of choice. Parameters $d$ and $d'$ are the number of feature channels.

We view a GCN layer as a local smoothing operation followed by an MLP, and simplify stacking K layers to $A^K H$ as in Wu et al. (2019a). In ReBraiD, every graph network layer aggregates information from each node's K-hop neighborhoods based on both brain structural connectivity and the latent adaptive adjacency matrix: namely we have both $A_i{}^K H^{(l)} W_K$ and $A_{i\_\text{adp}}{}^K H^{(l)} W_{K\_\text{adp}}$ for input $H^{(l)}$. We also gather different levels (from 0 to $K$) of neighbor information with concatenation. In other words, one graph convolution layer here corresponds to a small module that is equivalent to K simple GCN layers with residual connections. We can write our layer as:

$$H^{(l+1)} = \text{GNN}^{(l)}\left(H^{(l)}\right) = \text{MLP}\left[\text{Concat}_{k=1}^K \left(H^{(l)}, \text{ReLU}(A_i{}^k H^{(l)}), \text{ReLU}(A_{i\_\text{adp}}{}^k H^{(l)})\right)\right] \quad (2)$$

With the additional temporal dimension, $H^{(l)} \in \mathbb{R}^{N \times d \times t}$ in eq. (2), and $A_i \in \mathbb{R}^{N \times N}$ applies on $H^{(l)}$'s first two dimensions while multiplying. Outputs of different $\text{GNN}^{(l)}$ layers are parametrized and then skip connected with a summation. Since the temporal lengths of these outputs are different because of TCNs, max-pooling is used before each summation to make the lengths identical.

**(IV) Multi-resolution inner cluster smoothing.** While GNN layers can effectively passing information between neighboring nodes, long-range relationships among brain regions that neither appear in SC nor learned by latent $A_\text{adp}$ can be better captured using soft assignments similar to DIFFPOOL (Ying et al. (2018)). To generate the soft assignment tensor assigning $N$ nodes into $c$ clusters ($c$ chosen manually), we use $\text{GNN}_{pool}^{(l)}$ that obeys the same propagation rule as in eq. (2), followed by Softmax along $c$. This assignment is applied to the output of $\text{GNN}_{emb}^{(l)}$ which carries out the spatial embedding for the $l^{th}$ layer input $H^{(l)}$:

$$S^{(l)} = \text{Softmax}\left(\text{GNN}_{pool}^{(l)}\left(H^{(\ell)}\right), 1\right) \in \mathbb{R}^{N \times c \times t}$$
$$Z^{(l)} = \text{GNN}_{emb}^{(l)}\left(H^{(l)}\right) \in \mathbb{R}^{N \times d \times t} \quad (3)$$
$$\tilde{H}^{(l)} = S^{(l)\top} Z^{(l)} \in \mathbb{R}^{c \times d \times t}$$

The extra temporal dimension allows nodes to be assigned to heterogeneous clusters at different frames. We find that using coarsened $A_i^{(l+1)} = S^{(l)\top} A_i^{(l)} S^{(l)} \in \mathbb{R}^{c \times c}$ as the graph adjacency matrix leads to worse performance compared to using SC-generated $A_i$ and learned $A_{i\_\text{adp}}$ (comparison in section 3.1). In addition, if the number of nodes is changed, residual connections coming from the beginning of temporal-spatial blocks can not be used and this impacts overall performance. To continue use $A_i$ and $A_{i\_\text{adp}}$ as graph adjacency matrices and allow residual connections, we reverse-assign $\tilde{H}^{(l)}$ with assignment tensor obtained from applying Softmax on $S^{(l)\top}$ along $N$, so that the number of nodes is kept unchanged:

$$\tilde{S}^{(l)} = \text{Softmax}\left(S^{(l)\top}, 1\right) \in \mathbb{R}^{c \times N \times t}$$
$$H^{(\ell+1)} = \tilde{S}^{(l)\top} \tilde{H}^{(l)} \in \mathbb{R}^{N \times d \times t} \quad (4)$$

In fact, eqs. (3) and (4) perform signal smoothing on nodes within each soft-assigned cluster (appendix A.1.2 shows a toy example). With the bottleneck $c < N$, the model is forced to pick up latent community structures. This inner-cluster smoothing is carried out at different spatial resolutions: as the spatial receptive field increases with more graph layers, we decrease cluster number $c$ for the assignment operation.

## 2.3 ATTRIBUTION WITH IG (INTEGRATED GRADIENTS).

As one approach to model interpretability, *attribution* assigns credits to each part of the input, assessing how important they are to the final predictions. Wiltschko et al. (2020) gives an extensive comparison between different graph attribution approaches, in which IG (Sundararajan et al. (2017)) is top-performing and can be applied to trained models without extra alterations of the model structure. IG also has other desirable properties such as implementation invariance that other gradient methods are lacking. It is also more rigorous and accurate than obtaining explanations from attention weights or pooling matrices that span multiple feature channels. Intuitively, IG calculates how real inputs contribute differently compared to a selected baseline; it does so by aggregating model gradients at linearly interpolated inputs between the real and baseline inputs. For each sample, we

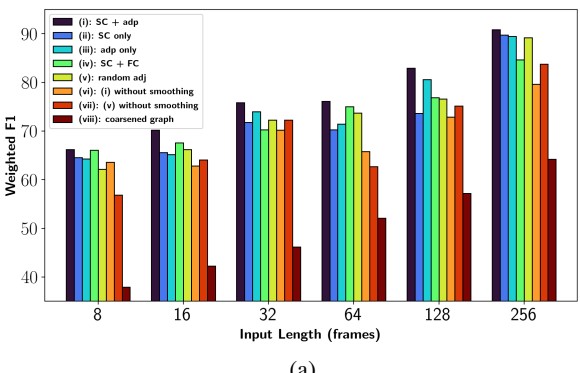 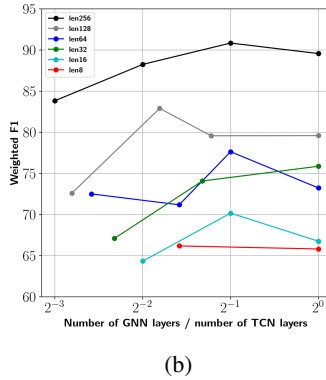

(a)            (b)

Figure 3: 3a: Ablation studies on different input length (please see table 3 in appendix for numerical values of weighted F1 under each setting); 3b: Choosing number of GNN to TCN layer ratio.

calculate attributions at each point of both input $A \in \mathbb{R}^{N \times N}$ and $X \in \mathbb{R}^{N \times T}$:

$$
\begin{aligned}
\mathrm{ATTR}_{A_{vw}} &= \left( A_{vw} - A'_{vw} \right) \times \sum_{m=1}^{M} \frac{\partial F\left( A_{\mathrm{Intrpl}}, X \right)}{\partial A_{\mathrm{Intrpl}_{vw}}} \times \frac{1}{M}, \quad A_{\mathrm{Intrpl}} = A' + \frac{m}{M} \times \left( A - A' \right) \\
\mathrm{ATTR}_{X_{vt}} &= \left( X_{vt} - X'_{vt} \right) \times \sum_{m=1}^{M} \frac{\partial F\left( A, X_{\mathrm{Intrpl}} \right)}{\partial X_{\mathrm{Intrpl}_{vt}}} \times \frac{1}{M}, \quad X_{\mathrm{Intrpl}} = X' + \frac{m}{M} \times \left( X - X' \right)
\end{aligned}
\tag{5}
$$

$F(A, X)$ here represents our signal classification model, $M$ is the step number when doing Riemann approximation of the path integral, and $A', X'$ are baselines of $A, X$ (see section 3.3 for more details). Note that eq. (5) calculates the attribution of one point on one sample. The process is repeated for every point of the input, so attributions have identical dimensions as inputs. To obtain brain region importance of a task, we aggregate attributions across multiple samples of that task.

## 3 EXPERIMENTS

We use fMRI signals from CRASH dataset (Lauharatanahirun et al. (2020)) for our experiments. The model classifies input fMRI into 6 tasks: resting state, visual working memory task (VWM), dynamic attention task (DYN), math task (MOD), dot probe task (DOT), and psychomotor vigilance task (PVT) (appendix A.2.1 has detailed task descriptions). We preprocess 4D voxel-level fMRI images into graph signals $\mathcal{G} = (A, X)$ by averaging voxel activities into regional signals with the 200-ROI cortical parcellation (voxel to region mapping) specified by Schaefer et al. (2018). We also standardize signals for each region and discard scan sessions with obvious abnormal spikes that may be caused by head movement, etc.. DWI scans are mapped into the same MNI152 coordinate and processed into adjacency matrices with the same parcellation as fMRI uses. Valid data contains 1940 scan sessions from 56 subjects, session length varies from 265 frames to 828 frames and TR (Repetition Time) is 0.91s. These 1940 scan sessions are separated into training, validation, and test sets with a ratio of 0.7-0.15-0.15. Hyperparameters including dropout rate, learning rate, and weight decay are chosen with grid search based on validation loss, and all results reported in this section are obtained from the test set. For each scan session, we use a sliding window to generate input sequences (in the following experiments $T \in \{8, 16, 32, 64, 128, 256\}$) and feed them to the model. To encode temporal and spatial information alternatively, we find stacking two TCN layers per one GNN layer leads to better performance most times (fig. 3b, see more on appendix A.2.3 (II)). Models are written in PyTorch, trained with Google Colab GPU runtimes, and 60 epochs are run for each experiment setting. Codes and data will be released upon acceptance.

### 3.1 MODEL COMPONENTS

**Graph adjacency matrices.** For each input sample $\mathcal{G}_i$, we test different options to provide graph adjacency matrices to the GNN layer. They include (i) our proposed method: using both adaptive adjacency matrix $A_{i\_\mathrm{adp}}$ and SC-induced $A_i$; (ii) only using $A_i$; (iii) only using $A_{i\_\mathrm{adp}}$; (iv) replacing $A_{i\_\mathrm{adp}}$ in setting i with $A_{i\_\mathrm{FC}}$ derived from functional connectivity; (v) only using random graph

adjacency matrices with the same level of sparsity as real $A$s. We set $h_{\text{adp}}$ to be 5 in eq. (1), which works better for our data than larger $h_{\text{adp}}$ choices. $K$ is set to 2 for eq. (2), meaning each GNN aggregates information from 2-hop neighbors based on the provided adjacency matrices. We evaluate our model with weighted F1 as the metric because of the imbalance among tasks. The results under different settings are reported in fig. 3a (and table 3 in appendix for numerical values).

From the results of setting (ii) plotted in fig. 3a, we see that removing the adaptive adjacency matrix impacts the performance differently at different input lengths: the gap peaks for signals of length 64 - 128, and becomes smaller for either shorter or longer sequences. This could suggest the existence of more distinct latent states of brain signals of this length that cannot be captured by structural connectivities. On the other hand, removing SC (setting (iii)) seems to have a more constant impact on the model performance, with shorter inputs more likely to see a slightly larger drop. In general, only using $A_{\text{adp}}$ leads to smaller performance drop than only using SC, indicating the effectiveness of $A_{\text{adp}}$ in capturing useful latent graph structures. More detailed studies in appendix A.2.3 shows $A_{\text{adp}}$ learns distinct representations not captured by $A$.

As mentioned in section 2, our motivation behind creating sample-level adaptive adjacency matrices is FC's highly dynamic nature. Therefore, for setting (iv), we test directly using adjacency matrices $A_{i\_\text{FC}}$ obtained from FC instead of the learned $A_{i\_\text{adp}}$. In particular, $A_{i\_\text{FC}} = \tilde{D}_{\text{FC}_i}^{-\frac{1}{2}} \tilde{\text{FC}}_i \tilde{D}_{\text{FC}_i}^{-\frac{1}{2}} \in \mathbb{R}^{200 \times 200}$, where $(\text{FC}_i)_{vw} = \text{corr}((X_i)_v, (X_i)_w)$, $\tilde{\text{FC}}_i = \text{FC}_i + I_N$ and $\tilde{D}_{\text{FC}_i} = \sum_w (\tilde{\text{FC}}_i)_{vw}$. Fig. 3a shows $A_{i\_\text{FC}}$ constantly underperforms $A_{i\_\text{adp}}$, except for being really close with length-8 inputs. Larger performance gaps are observed for longer inputs, where $\text{Corr}((X_i)_v, (X_i)_w)$ struggles to capture the changing dynamics in the inputs. This demonstrates that our input-based latent $A_{i\_\text{adp}}$ has better representation power than input-based FC. We also notice batch correlation coefficients calculation for $A_{i\_\text{FC}}$ results in a slower training speed than computing $A_{i\_\text{adp}}$.

An interesting result comes from setting (v), where we use randomly generated Erdős-Rényi graphs with the edge creation probability the same as averaged edge existence probability of $A$s. Its performance is at a similar level or even better than settings (ii) and (iii). Our hypothesis is the model can learn the latent graph structure out of randomness, and we will verify this hypothesis in section 3.3.

**Multi-resolution inner cluster smoothing.** To verify the capability of inner-cluster smoothing operation in capturing latent graph dynamics, we test the following settings: (vi) using our proposed model and inputs, except removing paralleled $\text{GNN}_{pool}$ and inner-cluster smoothing module; (vii) previous setting (v) but remove $\text{GNN}_{pool}$ and inner-cluster smoothing module; (viii) keep $\text{GNN}_{pool}$, but using coarsened graph instead of performing smoothing and increasing the node number back (essentially performing DIFFPOOL with an added temporal dimension). In this last setting, we hierarchically pool graph nodes until node number reaches 1, and we keep the total number of GNN layers the same as our other settings. Values of soft-assigned cluster number $c$ are chosen to be halved per smoothing module, starting from half of the graph nodes number, namely: 100, 50, 25, etc. for our experiments. Different choices of $c$ affect model converging rate, but only have minor impacts on the final performance (see appendix A.2.3 (III)). Results are reported in fig. 3a (and table 3 in appendix). Apart from these three settings, we also test adding pooling regularization terms (described in appendix A.1.3) into the loss function but this does not lead to much of a difference.

The above results demonstrate that both setting (vi) and (vii) outperforms (viii) by a large margin, indicating the importance of keeping the original node number when representing brain signals. In addition, all three settings underperform our proposed method, and they are mostly worse than changing graph adjacency matrices as in settings (ii)-(v): this shows inner-cluster smoothing module has a large impact in learning latent graph dynamics. We also find using adaptive adjacency matrices and inner cluster smoothing can stabilize training, making the model less prone to over-fitting and achieving close-to-best performance over a larger range of hyperparameters (see fig. 11).

## 3.2 MODEL COMPARISONS

In this section, we compare our model with the vanilla GCN from Kipf & Welling (2017), Chebyshev Graph Convolutional Gated Recurrent Unit (GConvGRU) from Seo et al. (2018), GraphSAGE from Hamilton et al. (2017), GAT V2 from Brody et al. (2021) and Graph Transformer as in Shi et al. (2021). To use them in our fMRI classificaiton problem, we directly take corresponding layers from

Table 1: Model comparisons under the 256 input length setting.

| Model | Accuracy (%) | Weighted F1 | Training time (s / epoch) |
|---|---|---|---|
| GCN (Kipf & Welling (2017)) | 41.53 | 42.84 | 713 |
| GAT V2 (Brody et al. (2021)) | 50.44 | 50.36 | 1142 |
| GConvGRU (Seo et al. (2018)) | 52.26 | 56.05 | 9886 |
| GraphSAGE (Hamilton et al. (2017)) | 61.84 | 61.87 | 1048 |
| Graph Transformer (Shi et al. (2021)) | 66.51 | 66.11 | 1890 |
| **ReBraiD** (proposed method: TCN + GNN) | **85.56** | **90.85** | 298 |
| ReBraiD (TCN only) | 72.44 | 71.98 | 119 |
| ReBraiD (TCN + CNN) | 75.89 | 75.79 | 124 |

PyTorch Geometric [2] and PyTorch Geometric Temporal [3] and construct the models similar to ours: four graph layers taking in both signals and adjacency matrices, followed by two linear layers along node axis and two linear layers for the final classification. We train baseline models with the same setting as our model: 256-frame inputs, Adam optimizer, cross entropy loss, and 60 epochs (all models well-converged). Grid search is used to optimize the rest of hyperparameters. We compare accuracy, weighted F1 score, and training time per epoch in table 1; we also plot our model and Graph Transformer's confusion matrices in appendix fig. 12.

We observe that our proposed method significantly outperforms the baseline graph models by a margin of 20 to 40 percent and has much less training time. This demonstrates our proposed model's effectiveness in capturing latent brain dynamics. For these baseline models, temporal content is used as features; the comparison shows separating them into different axis is more advantageous. This is further confirmed with models only having TCN layers: we test both removing GNN layers all together and replacing them with $1 \times 1$ CNN layers. Both outperform graph models that focuses on the spatial modeling aspect. Although temporal modeling is crucial, including the spatial information in its graph format as our proposed model can improve the performance much further.

### 3.3 INTERPRETATIONS WITH IG

In this section, we study the contributions of different brain ROIs and subnetworks defined by their functionalities. For the subnetwork definition, we choose to use the 17 networks specified in Thomas Yeo et al. (2011)) which has a mapping from our previous 200-ROI parcellation. See table 4 in appendix for all subnetwork names. We compute IG of a model trained on length-256 input signals because the model has higher performance with longer inputs, leading to more accurate attributions. To select baseline inputs, we follow the general baseline selection principle for attribution methods: when the model takes in a baseline input, it should produce a near-zero prediction and $\mathrm{Softmax}(\mathrm{outputs})$ should give each class about the same probability in a classification model. All-zero $A'$ and $X'$ can roughly achieve this for our model, so we choose them as our baseline inputs. For each task, the IG computation is done on 900 inputs to get an overall distribution.

**Temporal importance.** On the single input level, we can attribute which parts of the inputs in $\mathcal{G}_i$ are more important in predicting the target class by looking into $(\mathrm{ATTR}_X)_i$. This attribution map not only shows which brain regions contribute more but also reveals the important signal frames. One critical drawback of fMRI imaging is its low temporal resolution, but if we can know which part is more important, we can turn to more temporally fine-grained signals such as EEG to see if there are any special activities during that time. To confirm that the attributions we get are valid and consistent, we perform a sanity check of IG results on two overlapped inputs with an offset $\tau$: the first input is obtained from window $[t_0, t_0 + T]$ and the second is obtained from window $[t_0 + \tau, t_0 + \tau + T]$. Offset aligned results are shown in fig. 4a, and we can see the attributions agree with each other quite well.

**Spatial importance.** We examine the connection importance between brain ROIs by looking at $\mathrm{ATTR}_A$ (task-averaged $\mathrm{ATTR}_A$ are plotted in fig. 14 in appendix). In particular, columns in $\mathrm{ATTR}_A$ with higher average values are sender ROIs of high-contributing connections, which is what matters in the GNN operation. We also explore why using random graph adjacency matrices (setting (v) in section 3.1) can produce a similar result for length-256 inputs compared to using both SC-induced

---

[2]https://pytorch-geometric.readthedocs.io/
[3]https://pytorch-geometric-temporal.readthedocs.io/

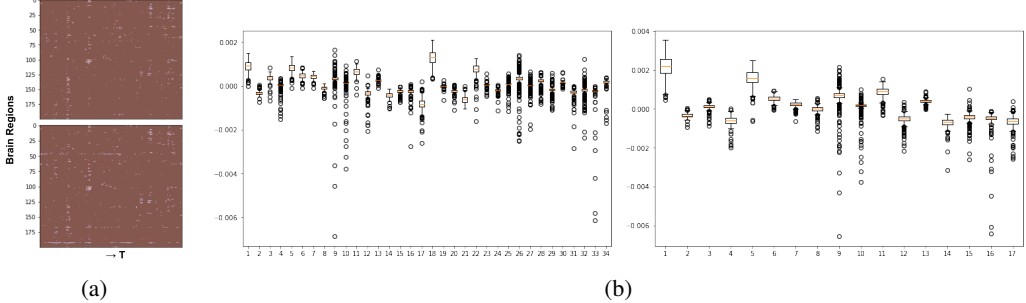

(a)                                              (b)

Figure 4: 4a: Temporal importance sanity check of IG results on two pieces of inputs with a large overlap period. Attribution maps are offset aligned; 4b: $\text{ATTR}_X$ distributions across brain regions for VWM task, where the upper one separates left and right hemispheres and the lower one combines them (e.g. LH_VisCent and RH_VisCent are combined to region VisCent). Refer to table 4 in appendix to see brain region names that the x-axis numbers represent.

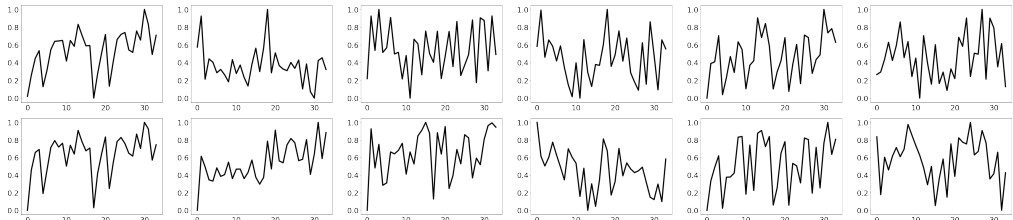

Figure 5: Column averages of task-averaged $\text{ATTR}_A$ (mapped into 34 subnetworks defined by the 17-network parcellation with left, right hemispheres). Top row is obtained from real SC induced $A$ and bottom rows is obtained from random SC induced $A_{\text{rand}}$. Attributions are normalized to $[0, 1]$.

$A_i$ and $A_{i\_\text{adp}}$ (setting (i)). By examining $\text{ATTR}_A$ under both settings (fig. 5), we see that the column averages of $\text{ATTR}_A$ under these two settings are similar for almost all tasks, meaning the model can learn where to pay attention to even using random adjacency matrix inputs. We credit this ability partially to multi-resolution inner cluster smoothing, as the performance would drop notably without it (setting (vii)). However, using ground truth SC not only gives us higher performance for shorter inputs but also provides the opportunity to better interpret brain region connections. We can directly use task-averaged $\text{ATTR}_A$ as the weighted adjacency matrix to plot edges between brain ROIs, just as in fig. 6. Important brain regions obtained from $\text{ATTR}_A$ mostly comply with the previous literature (see appendix A.2.5 for details).

$\text{ATTR}_X$ can also give us insights on spatial importance when the attribution maps are averaged or summed up along the temporal dimension. But it does so from another perspective: instead of showing important *structural connections* that support information passing, it reveals regions or subnetworks that are sources of the important *signals*. In fig. 4b, we plot the distribution of t-averaged and subnetwork-averaged (mapping 200 ROIs into 17 subnetworks) $\text{ATTR}_X$ during VWM task. We can see the clear dominance of VisCent, DorsAttnA, and ContA subnetworks (numbered as 1, 5, 11), indicating signals from these regions are useful for model to decide if the input is from VWM task. For the boxplots of other tasks and subnetwork rankings, please see fig. 16, table 5 and table 6 in appendix. More informative than the rankings is the distribution itself: even though VisCent, DorsAttnA, and ContA ranked top 3 for both resting state and VWM task for signal attributions, their relative importance and attribution distribution variances are totally different. In a sense, the distribution can act as a task fingerprint based on brain signal states.

We notice that signal-important ROIs are not necessarily the same as connection-important ROIs: top-ranked subnetworks for resting state are DefaultA and DefaultB by $\text{ATTR}_A$, and VisCent and DorsAttnA by $\text{ATTR}_X$; although they do coincide with each other for tasks like VMN. This disparity is reflected in fig. 6 as edge and node differences.

**Group, session, and region heterogeneity.** Average variances of attributions are very different across tasks, especially those of $\text{ATTR}_X$: VWM and DYN have much smaller attribution variances compared to other tasks. This can be caused by either task dynamics when certain tasks have more phase transitions and brain status changes, or/and group heterogeneity when individuals carry out

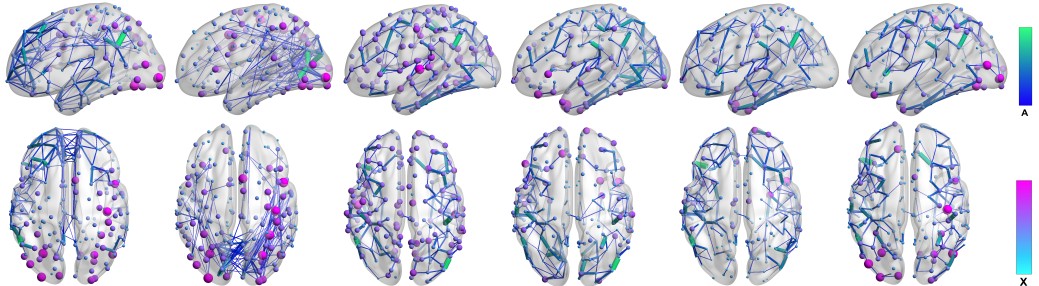

Figure 6: ROI attributions from ATTR$_A$ and ATTR$_X$. Tasks are: Resting, VWM, DYN, DOT, MOD, PVT from left to right. Edge color and width are based on task-averaged ATTR$_A \in \mathbb{R}^{200 \times 200}$, and nodes color and size are based on task and temporal-averaged ATTR$_X \in \mathbb{R}^{200}$. For the visualization purpose, only edges with highest attributions are kept to ensure sparsity being 0.009 (down from around 0.196). For ROI attributions based only on ATTR$_A$ where important sender ROIs are reflected by node sizes, please refer to fig. 13 in appendix.

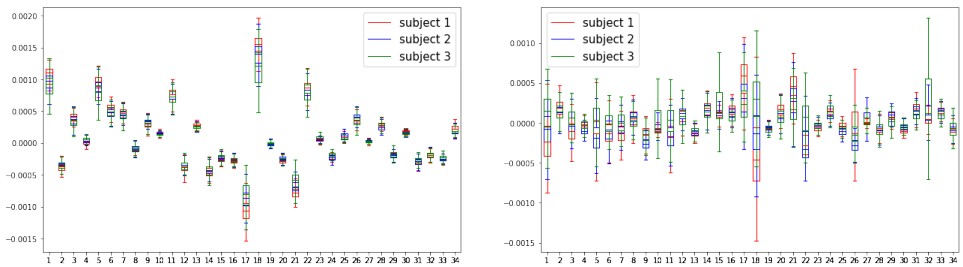

Figure 7: ATTR$_X$ distributions of 3 subjects doing VWM task (left) and MOD task (right). Outliers that go beyond $[Q1 - 1.5\,\mathrm{IQR}, Q3 + 1.5\,\mathrm{IQR}]$ are omitted. VWM has a much smaller average attribution variance than MOD.

certain tasks more differently than the others. We investigate the cause by looking into 3 subjects; each of them has multiple scan sessions for every task.

We report the following findings: (1) Even only aggregating attributions over a single subject's sessions, attribution variances of the other four tasks are still larger than VWM and DYN. The variance values are comparable to that of aggregating over many subjects. This means the large variances are not mainly due to group heterogeneity, rather some tasks having more states than others. (2) Apart from different task dynamics, there is still group heterogeneity. For tasks with more dynamics (high attribution variances), the group heterogeneity is also more obvious. We can see from fig. 7 that attributions for VMM are much more concentrated and universal across subjects than that of MOD. (3) Flexibility of different subnetworks varies: subnetworks that have small distribution IQR (Interquartile Range) of the same subject's different sessions will also be more consistent across subjects. One example is subnetwork 18 during MOD task has both higher within-subject IQR and larger across-subject differences compared to subnetwork 19. This indicates for a certain task, some subnetworks are more individual and flexible (may activate differently across $t$), while others are more universal and fixed. In summary, we can reveal both critical regions that a particular task must rely on, and regions that can characterize individual differences during tasks.

## 4 CONCLUSIONS

In this paper, we propose ReBraiD, a high-performing and efficient graph neural network model that embeds both structural and dynamic functional brain signals for task classifications. To better capture latent dynamics, we propose input-dependent adjacency matrix learning and inner-cluster smoothing at multiple resolutions. Apart from quantitative results showing ReBraiD's superiority in representing brain activities, we also leverage integrated gradients to attribute and interpret the importance of both spatial brain regions and temporal keyframes, as well as presenting heterogeneities among subnetworks, tasks, and individuals. These findings can potentially reveal new neural basis or biomarkers of tasks and brain disorders when combined with behavioral metrics, and enable more fine-grained temporal analysis around keyframes when combined with other imaging techniques.

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

# A APPENDIX

## A.1 MODELS

### A.1.1 DYNAMIC FUNCTIONAL CONNECTIVITIES.

Fig. 8 shows functional connectivities (FCs) among $N$ brain regions, where each FC $\in \mathbb{R}^{N \times N}$. The value at $FC_{ij}$ is calculated as the Pearson correlation coefficient between signals of brain region $i$ and region $j$. The figure shows 6 FCs calculated from 6 consecutive sliding windows within a same fMRI session, with signal window length being 30 and sliding stride being 30. From the figure, we can clearly tell that FCs are highly dynamic.

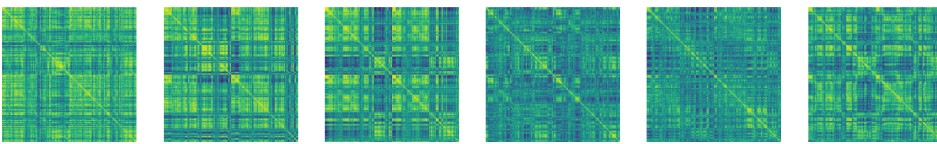

Figure 8: Dynamic functional connectivities.

### A.1.2 INNER-CLUSTER SMOOTHING TOY EXAMPLE.

Here we show a toy example demonstrating the inner-cluster smoothing module described in eqs. (3) and (4). Note that we will only show one time slice, and the same operation is done along every $t$: on a particular $t$, we have $Z \in \mathbb{R}^{N \times d}, S \in \mathbb{R}^{N \times c}$. We will use $N = 3, c = 2$ and node values $a, b, c \in \mathbb{R}^d$ for this toy example. In addition, this example is just to illustrate the concept behind the smoothing operation, and $\mathrm{Softmax}$ along the axis 1 is simplified as row normalization for a clearer presentation.

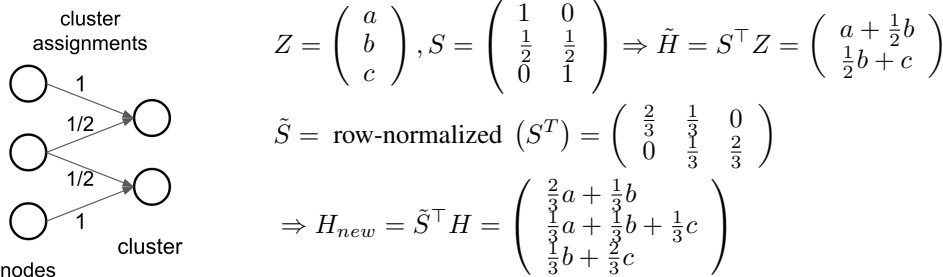

Figure 9: Inner-cluster smoothing toy example.

In this example, $1^{st}$ and $2^{nd}$ nodes are assigned to the first cluster, and $2^{nd}$ and $3^{rd}$ node are assigned to the second cluster. The final $H_{new}$ after our smoothing module will mingle the first two nodes' values, and the last two nodes' values (based on assignment weights) while keeping their original node number unchanged.

### A.1.3 REGULARIZATION TERMS FOR THE SOFT-ASSIGNMENT.

For each soft assignment matrix $S \in \mathbb{R}^{N \times c \times t}$ in eq. (3), we test three regularization terms:

- Similar to DIFFPOOL, for ensuring a more clearly defined node assignment, namely each node is only assigned to few clusters (the closer to one the better), we can minimize the entropy of single node assignments: $L_{E_1} = \frac{1}{c} \sum_{i=1}^{c} H(S_i)$, where $S_i$ is along $c$.
- To ensure a representation separation among nodes, meaning the assignment shouldn't assign all the nodes a same way, we maximize the entropy of node assignment *patterns* across all nodes: $L_{E_2} = -\frac{1}{c} \sum_{i=1}^{c} H(\sum_{j=1}^{n} S_{ij})$, where $j$ is along $n$ and $i$ is along $c$.

- To make assignment along temporal axis smoother, we penalize assignment variances within a small window $[\hat{t}, \hat{t}+\tau]$: $L_T = \frac{1}{t-\tau} \sum_{\hat{t}=0}^{t-\tau} \sigma(S_{[\hat{t}, \hat{t}+\tau]})$, where $\sigma$ represents standard deviation.

Together with cross entropy classification loss $L_{CE}$, the final loss function of the model will be:

$$L_{reg} = \alpha_1 L_{CE} + \alpha_2 L_{E_1} + \alpha_3 L_{E_2} + \alpha_4 L_T, \quad \sum_i \alpha_i = 1 \qquad (6)$$

## A.2 EXPERIMENTS

### A.2.1 TASK DESCRIPTIONS.

The following are task descriptions of CRASH (Cognitive Resilience and Sleep History) dataset:

**Resting state**: The subject simply lays in the scanner awake, with eyes open for 5 minutes.

**Visual working memory task (VWM)**: The subject is presented with a pattern of colored squares on a computer screen for a very brief period (100ms). After 1000ms, they are presented with a single square and must determine if it is the same or different color as the previously presented square at that location. Responses are made with a button press (Luck & Vogel (1997)).

**Dynamic Attention Task (DYN)**: Two streams of orientation gratings are presented to the left and right of fixation. Subjects monitor specified stream for a target (about 2 degree shift in orientation, clockwise or counter clockwise) that indicates whether the subject should continue to monitor the current stream (hold) or monitor the other stream (shift) and respond with a button press (Yantis et al. (2002)).

**Dot Probe Task (Faces) (DOT)**: On each trial, two faces are presented, one neutral and the other happy or angry for 500ms. Then, either of two simple symbols is presented at the position of either of the faces. The subject must make a forced choice discrimination of the symbol. Reaction time differences as a function of the valance for the preceding facial expression are calculated. There is increased variability of the bias with PTSD and fatigue (Sipos et al. (2014)).

**Math task (MOD)**: Subjects perform a modular math computation every 8 seconds and respond with a yes or no button press. The object of modular arithmetic is to judge the validity of problems such as 51=19(mod 4). One way to solve it is to subtract the middle number from the first number (i.e., 51–19) and then divide this difference is by the last number (32/4). If the dividend is a whole number, the answer is "true." Otherwise the answer is false (Mattarella-Micke et al. (2011)).

**Psychomotor vigilance task (PVT)**: The subject monitors the outline of a red circle on a computer screen for 10 minutes, and whenever a counter clockwise red sweep begins, they press a button as fast as possible. Subjects are provided with response time feedback. The experimenter records response latencies (Loh et al. (2004)).

### A.2.2 DATA DETAILS

After discarding scan sessions with abnormal spikes that may be caused by head movements, the valid session details for different tasks are listed in table 2.

Table 2: fMRI scan details for six tasks.

| Tasks | Resting | VWM | DYN | DOT | MOD | PVT | (Total) |
|---|---|---|---|---|---|---|---|
| Valid scan sessions | 209 | 514 | 767 | 155 | 138 | 157 | 1940 |
| Frames / Scan | 321 | 300 | 265 | 798 | 828 | 680 | / |

### A.2.3 ABLATION STUDIES

Numerical values of fig. 3a are reported in table 3. Training time ranges from 51 seconds / epoch for length-8 inputs to 298 seconds / epoch for length-256 inputs. Although the model is trained for 60 epochs in all experiments, it converges to a relatively stable loss level within 20 epochs.

Table 3: Weighted F1 of ablation study settings.

| Input length (frames) | 8 | 16 | 32 | 64 | 128 | 256 |
|---|---|---|---|---|---|---|
| (i): SC + adp | **66.19** | **70.18** | **75.87** | **76.14** | **82.91** | **90.85** |
| (ii): SC only | 64.54 | 65.58 | 71.79 | 70.31 | 73.63 | 89.79 |
| (iii): adp only | 64.32 | 65.20 | 74.01 | 71.42 | 80.63 | 89.46 |
| (iv): SC + FC | 66.10 | 67.58 | 70.26 | 75.02 | 76.91 | 84.68 |
| (v): random adj | 62.17 | 66.25 | 72.30 | 73.72 | 76.58 | 89.22 |
| (vi): (i) without smoothing | 63.57 | 62.82 | 70.19 | 65.82 | 72.91 | 79.65 |
| (vii): (v) without smoothing | 56.88 | 64.08 | 72.27 | 62.72 | 75.16 | 83.75 |
| (viii): coarsened graph | 37.92 | 42.23 | 46.18 | 52.12 | 57.17 | 64.25 |

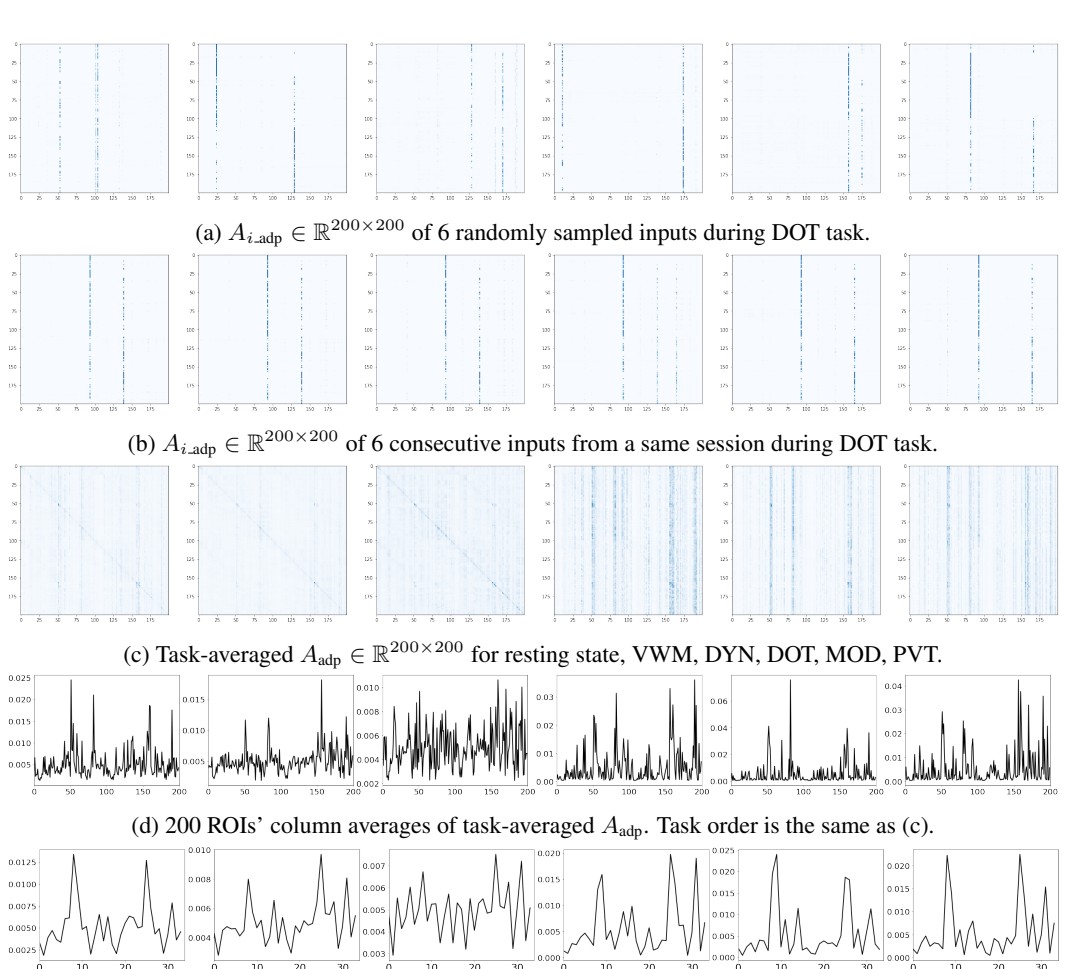

(a) $A_{i\_\mathrm{adp}} \in \mathbb{R}^{200 \times 200}$ of 6 randomly sampled inputs during DOT task.

(b) $A_{i\_\mathrm{adp}} \in \mathbb{R}^{200 \times 200}$ of 6 consecutive inputs from a same session during DOT task.

(c) Task-averaged $A_{\mathrm{adp}} \in \mathbb{R}^{200 \times 200}$ for resting state, VWM, DYN, DOT, MOD, PVT.

(d) 200 ROIs' column averages of task-averaged $A_{\mathrm{adp}}$. Task order is the same as (c).

(e) 34 (17 with LR) subnetworks' column averages of task-averaged $A_{\mathrm{adp}}$. Task order is the same as (c).

Figure 10: Learned latent adaptive adjacency matrices $A_{\mathrm{adp}}$.

**(I) Latent adaptive adjacency matrix $A_{\mathbf{adp}}$.** As we mentioned in section 3.1, latent $A_{\mathrm{adp}}$ can complement the task- and temporal-fixed $A$. We will now show that the learned $A_{i\_\mathrm{adp}}$ is sparse for each sample, has evident task-based patterns, and differs from what $A_i$ can provide: fig. 10 shows visualizations of latent $A_{\mathrm{adp}}$, which we can tell is quite sparse as in fig. 10a: each input only gets few important columns (information providing nodes in GNN), and they vary from one sample to another, indicating $A_{\mathrm{adp}}$'s ability to adapt to changing inputs within a same task. However, when we look into inputs (not shuffled) generated by consecutive sliding windows from a same scan session as in fig. 10b, we can see the latent structures appear to be in a smooth transition. In addition, when averaged across many samples for each task, undeniable task-based patterns emerge, as in figs. 10c

to 10e. But the task-average patterns are also different from what we saw from $\text{Attr}_A$ in fig. 5, suggesting $A_{\text{adp}}$ is embedding dynamics that are not captured by $A$. One interesting phenomenon in fig. 10e is that across all tasks, LimbicB_OFC, Default_B, and LimbicA_TempPole are always among the most important subnetworks appeared in the latent $A_{\text{adp}}$. Further exploration is needed for explaining the case.

**(II) Number of GNN layers.** The total number of temporal layers depends on the input signal length since each strided TCN layer reduces the temporal length by a factor of 2: if the input length is $2^i$, there need to be $i$ temporal layers. But *is alternating every TCN with GNN the best strategy, or do we only need to follow one GNN after a few TCNs*? We study this question with different input lengths.

Model weighted F1 are plotted in fig. 3b for all possible GNN to total TCN ratios (e.g. length-256 inputs requires 8 TCN layers. The possible ratios are $\frac{1}{8}, \frac{1}{4}, \frac{1}{2}, 1$ since we can insert one GNN per 8, 4, 2, 1 TCN layers). The figure shows alternating every layer rarely yields the highest performance and the best ratio lines around one GNN per two TCN layers for our dataset. We repeat the experiment for $K = 1, 3$ (in eq. (2)) to rule out the possibility that this result is related to how many neighbors one GNN layer can reach; we find they have roughly the same pattern as the $K = 2$ case. Our hypothesis is that lower GNN to TCN ratio does not capture enough spatial context, while higher ones might be overfitting. We leave exploring the relationship between this ratio and nodes number $N$ to a future study.

The best GNN to TCN ratio also depends on whether model incorporates latent adjacency matrices or not: without $A_{\text{adp}}$, length-128 signals achieves its relative best (among all ratios) when having one GNN per two TCNs, but it only needs one GNN per three TCNs if using $A_{\text{adp}}$. This shows learning latent structures $A_{\text{adp}}$ not only improves overall model accuracy but can also reduce model parameters, thus complexity, in achieving the relative best results.

**(III) Effects of soft-assignment cluster numbers.** During our experiments, we find as long as the smoothing module is used, the final performance will be close to each other, only the convergence rates are different. Fig. 11b shows how validation loss converges with different $c$ settings and when there is no smoothing module used. From it, we can see halving numbers (100-50-25-12) is most helpful and we use it for our other experiments; decreasing numbers (160-120-80-40) or all larger numbers (all 100) works better than increasing numbers (12-25-50-100) or all smaller numbers (all 12). Using the inner-cluster smoothing module, all cluster number settings converges to around 0.23 at their smallest when trained for 60 epochs and have test weighted F1 from 89.47 (model with 12-25-50-100) to 90.85 (model with 100-50-25-12). On the contrary, if no smoothing module is used, the model overfits easily and the validation loss can only reach about 0.4 before going up (with the best set of learning rate and weight decay parameters found with grid search). It is understandable that the model is prone to overfitting given the complexity of GNN and the relatively small dataset size. Our added inner-cluster smoothing module seem to effectively countering the effect and brings the loss down further and stabler.

### A.2.4 MODEL COMPARISONS.

We plot confusion matrices of our model, model from ablation study setting (viii), and the best performing baseline in fig. 12. Misclassification pairs clustered as the first three tasks (resting, VWM, DYN) and the latter three (DOT, MOD, PVT). Shown confusion matrices are from models trained on 256-frame inputs. We note that these misclassification pairs may be different for models trained on other input lengths (like 128-frame, etc.).

### A.2.5 IG $\text{Attr}_A$ AND $\text{Attr}_X$

See fig. 13 for the visualization of important ROIs based on $\text{Attr}_A$, and fig. 14 for task-averaged $\text{Attr}_A$ under real and random SC settings. Many discriminatory regions obtained from $\text{Attr}_A$ complies with the literature:
**For resting state**: the top attributed ROIs belong to the default mode network, which is regarded salient during the resting state (Raichle, 2015).
**For VWM**: the dominant attributions are from visual regions and posterior parietal regions, which complies with Todd & Marois (2004).
**For DYN**: attributions from our model suggest regions along cingulate gyrus (defaultA-

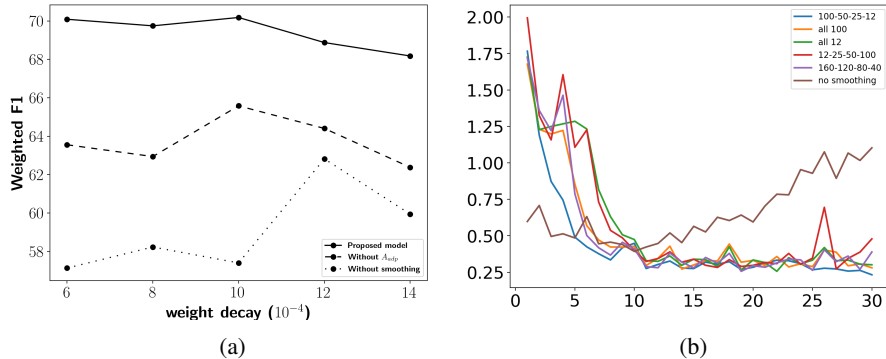

(a)                                    (b)

Figure 11: 11a: adding inner cluster smoothing or input-dependent adaptive adjacency matrix makes the model more stable across various learning rates. 11b: Validation loss v.s. training epochs. Input length is 256 and four smoothing modules are used. Legends are the soft-assignment cluster numbers of the four smoothing modules. Our other experiments are using decreasing cluster numbers that halved each module, corresponding to the 100-50-25-12 here.

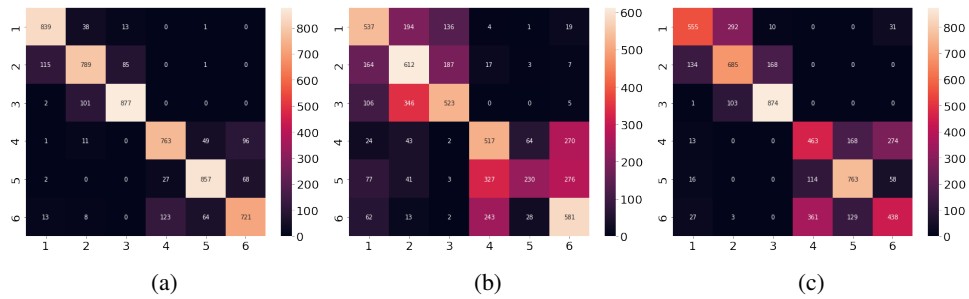

(a)                              (b)                              (c)

Figure 12: Confusion matrices of: (12a) ReBraiD (our proposed model), (12b) model with coarsened graph (setting (viii)), (12c) Graph Transformer (best-performing baseline). Tasks are 1-Resting, 2-VWM, 3-DYN, 4-DOT, 5-MOD, 6-PVT.

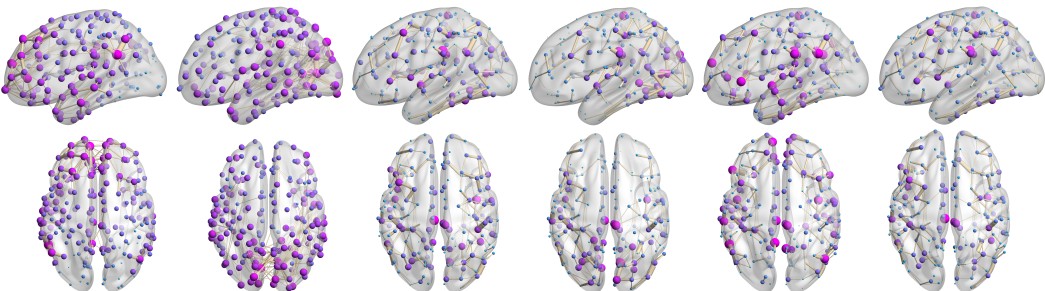

Figure 13: Important ROIs based on $\text{Attr}_A$. Tasks are: Resting, VWM, DYN, DOT, MOD, PVT from left to right. Node sizes are based on column sums of $\text{Attr}_A \in \mathbb{R}^{200 \times 200}$ and edge width are direcly based on $\text{Attr}_A$. For the visualization purpose, only edges with highest attributions are kept to ensure sparsity being 0.009 (down from around 0.196).

SalValAttnB-ContA-ContC-defaultC), as well as peripheral visual and somatomotor regions. Literature suggests anterior cingulate cortex (ACC) to be active (Kim et al., 2016) and posterior cingulate cortex (PCC) to be inactive (Leech & Sharp, 2014) during visual attention tasks. This means both regions provide discriminative information about classifying DYN states, which is what our attribution method is voting for.

**For DOT**: important ROIs from our analysis are located in control networks, in particular both ACC and PCC, as well as in the peripheral visual system. In the literature, dorsal and rostral regions of the ACC are proved to be involved with dot-probe performance (Carlson et al., 2012; 2013).

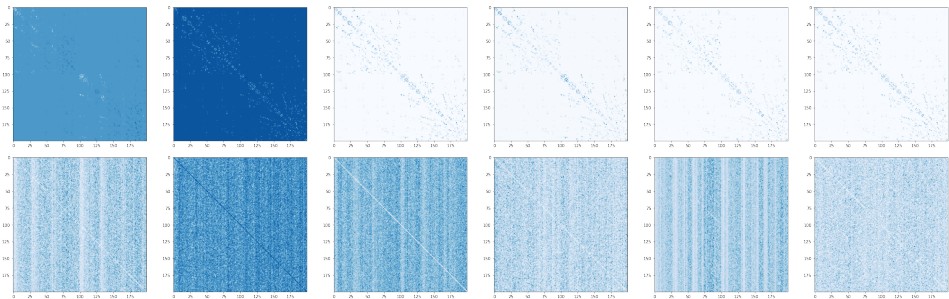

Figure 14: Task-averaged $\text{ATTR}_A \in \mathbb{R}^{200 \times 200}$. The top row is obtained from real SC induced $A$, and the bottom row is obtained from random SC induced $A_{\text{rand}}$.

**For MOD**: our important ROIs are mostly in temporal-parietal regions and default mode network (anatomically fronto-parietal), and literature suggests similar regions: parietal (Grabner et al., 2011) and prefrontal (Friedrich & Friederici, 2013).

**For PVT**: our top attributed ROIs belong to control networks, attention networks, and somatomotor regions. Similar as stated in Drummond et al. (2005), where both attention and motor systems are considered important.

To view brain regions in the 17-networks setting instead of 200-ROI parcellations, table 5 has the subnetwork rankings based on column-average $\text{Attr}_A$ and table 6 has the subnetwork rankings based on temporal-averaged $\text{Attr}_X$. For a visualization of the 17-network parcellation, please refer to fig. 15.

See fig. 16 for the complete attribution distributions for every task based on temporal-averaged $\text{Attr}_X$. Corresponding brain regions of a certain number are listed in table 4. For boxplots showing 17 regions, we combine LH and RH for their common network.

Table 4: Brain subnetworks in the 17-network definition.

| | | | |
|---|---|---|---|
| 1 | LH_VisCent | 18 | RH_VisCent |
| 2 | LH_VisPeri | 19 | RH_VisPeri |
| 3 | LH_SomMotA | 20 | RH_SomMotA |
| 4 | LH_SomMotB | 21 | RH_SomMotB |
| 5 | LH_DorsAttnA | 22 | RH_DorsAttnA |
| 6 | LH_DorsAttnB | 23 | RH_DorsAttnB |
| 7 | LH_SalVentAttnA | 24 | RH_SalVentAttnA |
| 8 | LH_SalVentAttnB | 25 | RH_SalVentAttnB |
| 9 | LH_LimbicB_OFC | 26 | RH_LimbicB_OFC |
| 10 | LH_LimbicA_TempPole | 27 | RH_LimbicA_TempPole |
| 11 | LH_ContA | 28 | RH_ContA |
| 12 | LH_ContB | 29 | RH_ContB |
| 13 | LH_ContC | 30 | RH_ContC |
| 14 | LH_DefaultA | 31 | RH_DefaultA |
| 15 | LH_DefaultB | 32 | RH_DefaultB |
| 16 | LH_DefaultC | 33 | RH_DefaultC |
| 17 | LH_TempPar | 34 | RH_TempPar |

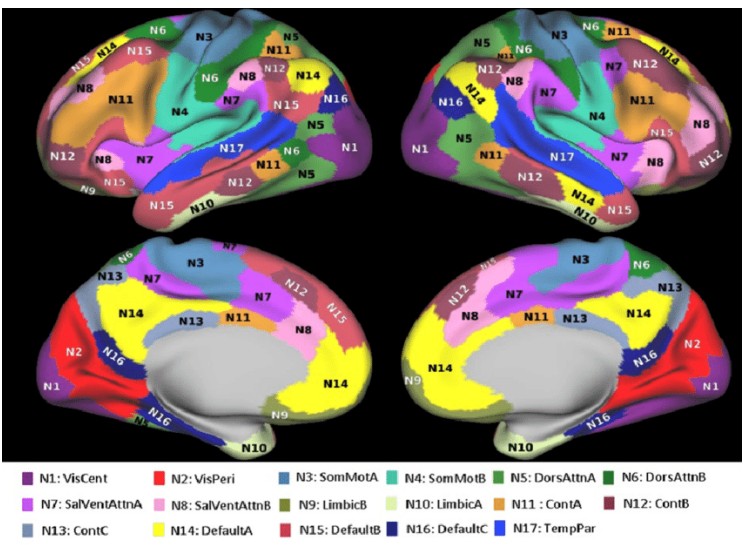

Figure 15: Network parcellation of Yeo's 17-networks. Figure is from Kahali et al. (2021)

Table 5: $\text{Attr}_A$: Top 10 brain subnetworks initiate important connections during different tasks (fig. 13 provides a ROI-based visualization).

| | rest | VWM | DYN | DOT | MOD | PVT |
|---|---|---|---|---|---|---|
| 1 | RH_DefaultA | RH_VisPeri | LH_SomMotB | RH_VisPeri | RH_DefaultA | RH_ContA |
| 2 | LH_DefaultA | LH_VisPeri | RH_DefaultC | LH_VisPeri | LH_DefaultA | RH_SalVentAttnA |
| 3 | RH_DefaultB | LH_VisCent | LH_VisPeri | RH_ContC | LH_DefaultC | RH_ContC |
| 4 | RH_ContB | RH_VisCent | LH_SalVentAttnA | RH_DorsAttnA | RH_DefaultC | LH_SalVentAttnA |
| 5 | RH_LimbicB_OFC | LH_DefaultC | RH_ContC | RH_SalVentAttnA | RH_DefaultB | RH_DefaultA |
| 6 | RH_SalVentAttnB | RH_SomMotB | RH_ContA | LH_ContC | RH_LimbicB_OFC | LH_ContC |
| 7 | RH_SomMotB | RH_DefaultC | RH_DefaultA | RH_DefaultC | LH_SomMotB | RH_DorsAttnA |
| 8 | RH_TempPar | LH_SomMotB | RH_SalVentAttnA | LH_SomMotB | RH_LimbicA_TempPole | LH_LimbicB_OFC |
| 9 | LH_DefaultB | LH_LimbicA_TempPole | LH_DefaultC | RH_ContA | LH_DefaultB | LH_SomMotB |
| 10 | RH_SalVentAttnA | RH_LimbicA_TempPole | RH_VisPeri | RH_VisCent | RH_SomMotB | RH_DefaultC |

Table 6: $\text{Attr}_X$: Top 10 brain subnetworks that are sources of the important signals during different tasks (fig. 6 provides a ROI-based visualization).

| | rest | VWM | DYN | DOT | MOD | PVT |
|---|---|---|---|---|---|---|
| 1 | RH_VisCent | RH_VisCent | LH_TempPar | RH_LimbicB_OFC | LH_TempPar | RH_VisCent |
| 2 | LH_VisCent | LH_VisCent | RH_SomMotB | LH_LimbicA_TempPole | LH_LimbicA_TempPole | LH_DorsAttnA |
| 3 | LH_DorsAttnA | LH_DorsAttnA | LH_VisPeri | RH_DefaultB | RH_SomMotB | LH_VisCent |
| 4 | RH_DorsAttnA | RH_DorsAttnA | LH_DefaultA | LH_TempPar | LH_DefaultB | RH_DorsAttnA |
| 5 | LH_ContA | LH_ContA | LH_DefaultC | RH_LimbicA_TempPole | RH_DefaultA | RH_LimbicB_OFC |
| 6 | LH_DorsAttnB | LH_DorsAttnB | RH_DefaultC | RH_TempPar | RH_LimbicA_TempPole | LH_ContA |
| 7 | LH_SomMotA | LH_SalVentAttnA | RH_SomMotA | LH_DefaultB | LH_LimbicB_OFC | RH_LimbicA_TempPole |
| 8 | LH_SalVentAttnA | LH_SomMotA | LH_ContB | LH_LimbicB_OFC | RH_DefaultB | LH_SomMotA |
| 9 | RH_ContA | RH_LimbicB_OFC | RH_SalVentAttnA | RH_SomMotB | LH_ContB | LH_LimbicB_OFC |
| 10 | LH_LimbicB_OFC | LH_LimbicB_OFC | LH_DefaultB | LH_DefaultA | LH_DefaultA | LH_DorsAttnB |

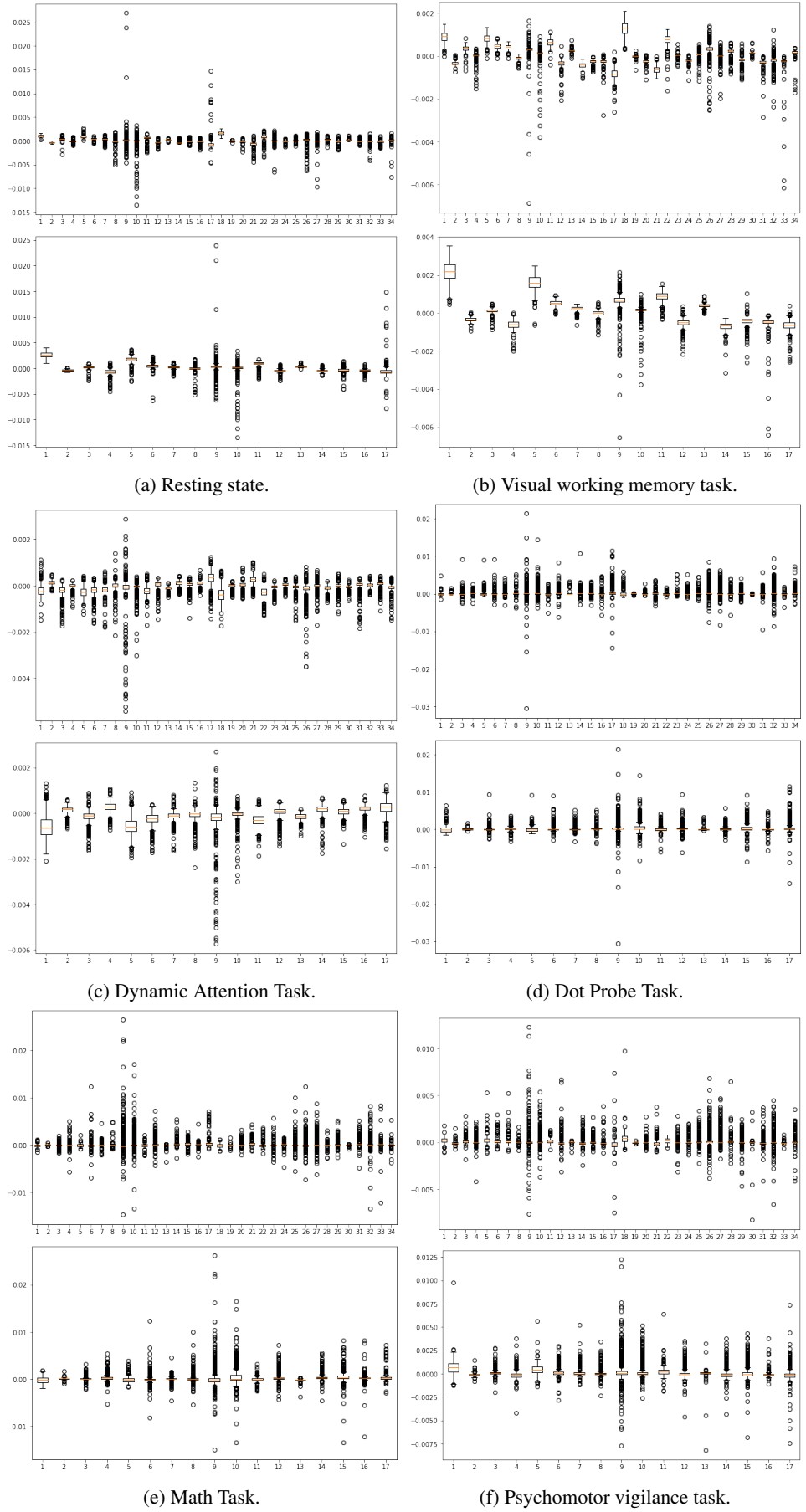

(a) Resting state.

(b) Visual working memory task.

(c) Dynamic Attention Task.

(d) Dot Probe Task.

(e) Math Task.

(f) Psychomotor vigilance task.

Figure 16: Brain subnetwork attribution distributions from $\mathrm{Attr}_X$.

### A.2.6 ATTRIBUTION REPRODUCIBILITY

In order to be used for downstream tasks, the extracted important regions should be reproducible across different initializations. Since the overall problem is non-convex, we test the reproducibility empirically with two models trained on two data splits, both initialized randomly and taking in length 256 inputs.

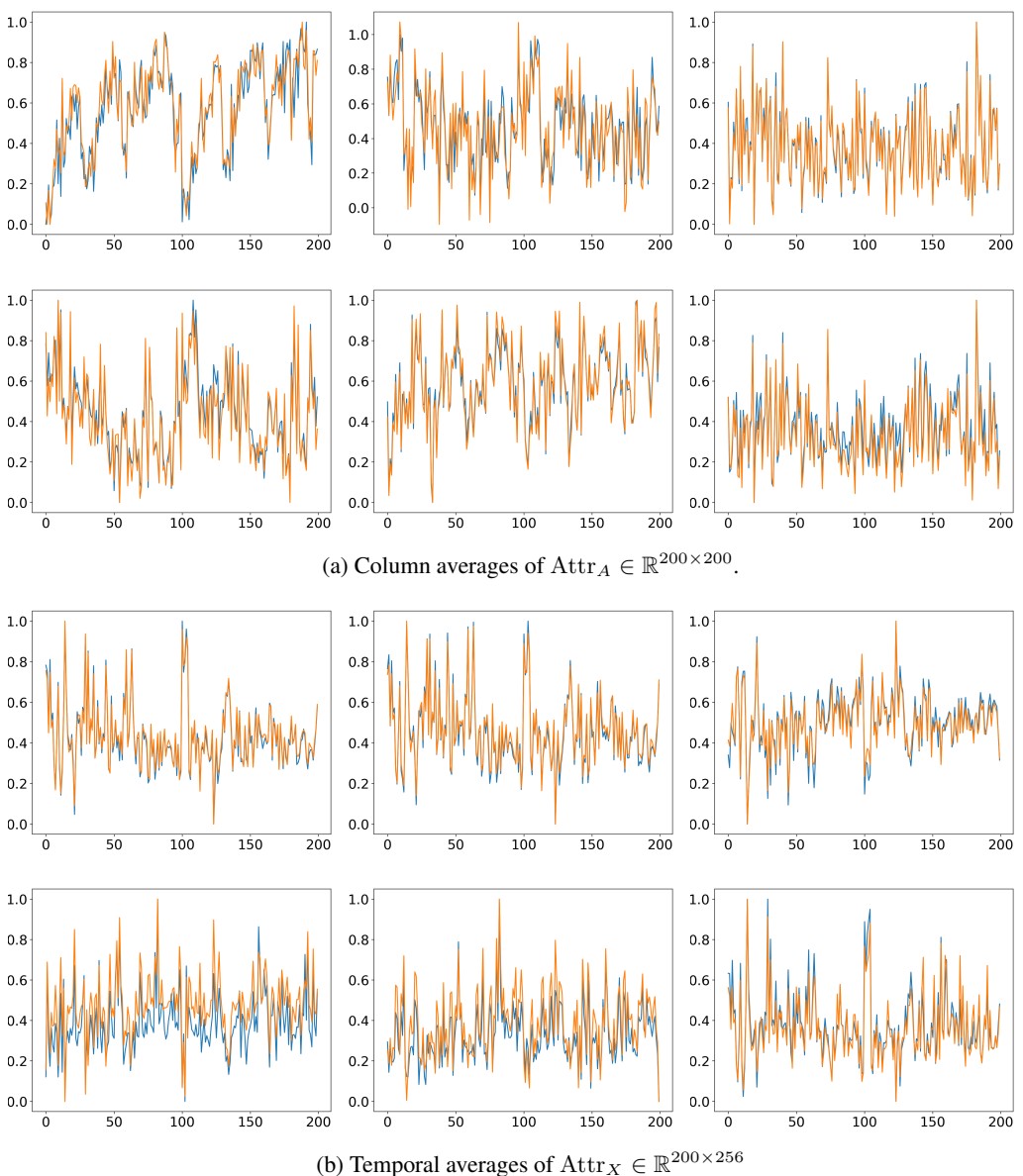

(a) Column averages of $\mathrm{Attr}_A \in \mathbb{R}^{200 \times 200}$.

(b) Temporal averages of $\mathrm{Attr}_X \in \mathbb{R}^{200 \times 256}$.

Figure 17: Reproducibility validation: attributions obtained from two models trained on different data split and initialized randomly. Attribution values are normalized to $[0, 1]$ and the larger the attribution value, the stronger indicating power that ROI has for a certain task. Tasks are Resting, VWM, DYN, DOT, MOD, and PVT.

### A.2.7    SIMULATION STUDY

To validate interpretation results, we perform simulation studies with know ground truth. For generating graph signals, we first define graph structures. All graphs are generated with stochastic block model (SBM) using a same community structure (200 nodes, 10 communities), but each graph has its own adjacency matrix. This mimics brain structures in that samples share similar community structures but have different structural connectivities. Fig 18a shows a typical adjacency matrix of a synthetic graph. All adjacency matrices are binary. Time-series on each node are then generated with codes adapted from this repo [4]. In particular, the value at each time step of each node is a small temporal Gaussian random noise plus signals from neighbors (a small Gaussian spatial noise is added to the adjacency matrix).

**Simulation (I)** We create two classes for this simulation: in class one, only the first three communities (nodes 1 - 60) generate small temporal noises and other nodes are only affected by neighbors; in class two, only the last three communities (nodes 141 - 200) generate small temporal noises and other nodes are only affected by neighbors. We visualize task aggregated $\text{Attr}_X$ and $A_{\text{adp}}$ and in figs. 18b and 18c. The signal importance differences are relatively well reflected in $\text{Attr}_X$. For the generated series, signals are more important in node 1 - 60 for class 1 and 141 - 200 for class 2: $A_{\text{adp}}$ catches this up and help propagating signals in these regions better. We notice that $\text{Attr}_A$ are mostly random and no obvious patterns are shown. This also reflects the graph signal generation: when aggregating information from neighbors, all connected edges are weighted the same (binary), thus the connections not really affect generated signals. To see the opposite, we perform another study below.

**Simulation (II)** We again create two classes for the simulation: in class one, connections from nodes 61 - 100 are strengthened; in class two, connections from nodes 101 - 140 are strengthened. The weights of strengthened edges are increased from 1 to 5 during signal generation, but model still takes in binary adjacency matrices as inputs (processed as mentioned in section 2.1 before feeding to the model). We visualize task aggregated $A_{\text{adp}}$ and $\text{Attr}_A$ in figs. 18d and 18e. We can see this time the connection differences are reflected in $\text{Attr}_A$. Signals in node 61 - 100 for class 1 or 101 -140 for class 2 are less important because they can be sent out by stronger connections: this results in smaller values for corresponding columns in $A_{\text{adp}}$. Combined with previous simulation, regions that are strong signal senders and connections from them are weak or not reflected in graph adjacency matrices tend to have higher $A_{\text{adp}}$ values. In other words, $A_{\text{adp}}$ complements both signals and connections to encode latent dynamics, while attributions obtained from IG are better at interpreting the modalities separately.

---

[4]https://github.com/alelab-upenn/graph-neural-networks

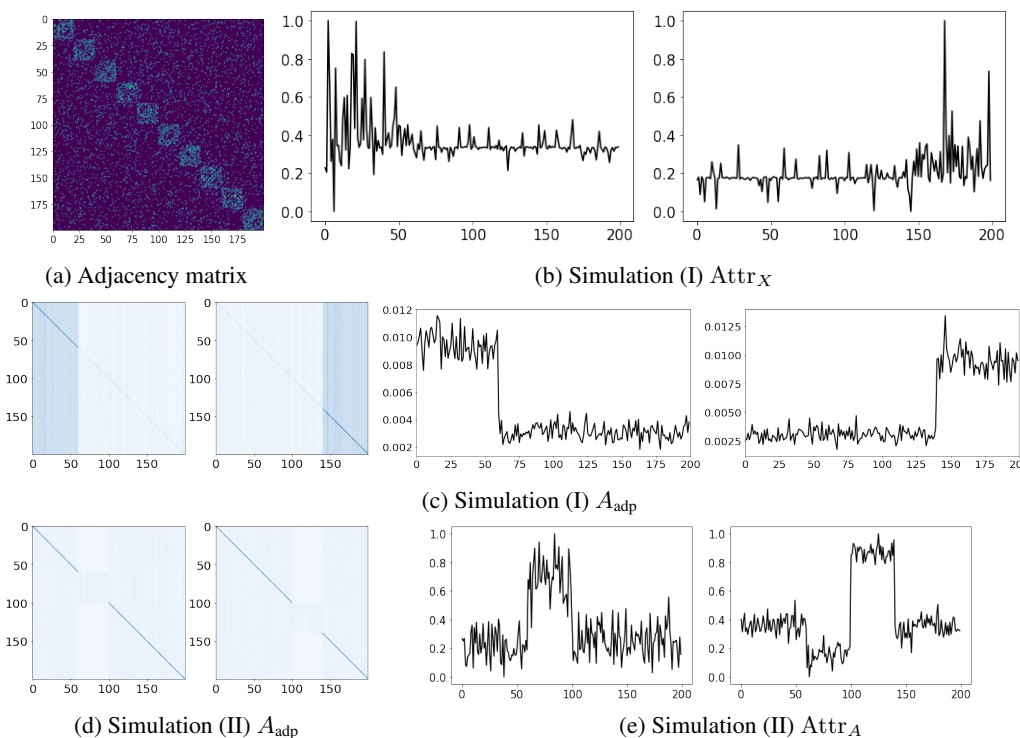

Figure 18: 18a: A typical adjacency matrix for simulated graph signals. 18b: Task averaged $\text{Attr}_X$ of simulation (I). Attribution values are normalized. 18c: Task averaged $A_{\text{adp}}$ of simulation (I) and its entry averages per column. 18d: Task averaged $A_{\text{adp}}$ of simulation (II). 18e: Task averaged $\text{Attr}_A$ of simulation (II). Attribution values are normalized.

