# OpenReview forum: "Deep Representations for Time-varying Brain Datasets"
_ICLR.cc/2022/Conference — ICLR 2022 Submitted_

### Official Review · Reviewer_Whpw · 2021-10-25

**Correctness:** 3
**Technical Novelty And Significance:** 3
**Empirical Novelty And Significance:** 2
**Recommendation:** 5
**Confidence:** 4

**Main Review:**

Pros:
1. The method in this article has made many improvements to graph neural networks. These improvements are to capture the spatial information and temporal information in brain dynamics. These improvements have also been verified in experiments to learn better temporal and spatial importance sanity.

2. The author uses the method of graph neural network to learn spatial information and temporal information. This new proposal may be extended to different fields. Not only does it have the effect of improving the brain image, but it may also be useful for general computer vision in the future. . So the author's innovation in the method is quite meaningful.

 Cons:
1. In the experiment, the author compared with other different graph neural network methods. In table 1, the accuracy of this method is about 20% higher than other methods. These results are quite confusing to me, and the GCN method only has 41% accuracy. I think the baseline method compared by the author may be too poor. It is more appropriate to compare it with other methods that are not graph neural networks. The ones in Table 1 The result is still unable to prove the effectiveness of the method.

2. The improvement of several methods on graph neural network in this paper is mainly based on some previous work. But the main contribution is to make some methodological adjustments for brain imaging, which greatly reduces the innovation point of this article. Therefore, the overall innovation of the article still needs to be improved.

**Summary Of The Paper:**

This paper proposes a method of graph neural network to jointly models dynamic functional signals and structural connectivities, to learn a good deep representation of brain dynamics. In this article, the author proposes some improved methods for graph neural networks, such as learning sample-level latent graph structures, temporal convolutional network and multi-resolution inner cluster smoothing methods to better learn insightful interpretations of brain dynamics. In the experiment, the author verified the effectiveness of the method in this paper on the fMRI signal data.

**Summary Of The Review:**

This article improves the network method of graph neural network to learn a better representation for brain dynamics in the application of brain images. I prefer to reject this article because this article has made some small improvements on GNN, which is more like to combine them in different ways to improve the overall effect. In addition, the baseline effects compared by the author in the experiment are relatively weak, and the effectiveness of the method cannot be demonstrated.

---

> ### Author Response · Authors · 2021-11-22
> **Response**
>
> Thank you for your comments. Regarding your concerns:
> - We did confine our baseline search to graph models since we identified our problem in the realm of network neuroscience, where brain regions are treated as graph nodes. To search for baseline models, We went through all models in PyTorch Geometric (PyG) library and its Temporal (PyG-T) extension. They contain the most up-to-date and well-organized open-source graph neural network model implementations. For each selected baseline, the constructed model contains 4 corresponding graph layers for encoding (ours also has 4 graph layers), and the same decoder as ours (2 pooling layers on node axis, and 2 linear layers for classification), hyper-parameters are tuned with grid search.
> We are aware of the large performance gap. We think the most crucial reason is PyG treats temporal signals as feature vectors instead of putting them into a separated temporal dimension. Without sequence modeling on the temporal dimension, even the recently published graph attention models (GAT-v2 and graph Transformer) cannot give a good performance. As for PyG-T, almost all models in it assume one graph for the inputs (application scenarios are traffic network forecasting, link predictions, etc.), but we need to feed different adjacency matrices for every sample. Out of them, we were able to choose one model (GConvGRU) that supports different adjacency matrices, but it didn’t give a satisfactory result. Given this, we think our proposed sample-level adaptive adjacency matrix learning and multi-resolution inner cluster smoothing are useful in improving performance. This is also demonstrated with ablation studies.
> In the revision, we also add additional TCN models into the baseline, which perform better than GCN models. However, adding graph structure into signal encoding as our proposed method not only improves the performance further but also leads to better interpretability of brain networks.
> - We think our contributions are the following:
> Most of the previous spatial-temporal GCNs are applied to traffic predictions and learn a universal adjacency matrix. Instead, we propose sample-level adaptive adjacency matrix learning based on input snippets, and this can better capture changing input dynamics under different time or task states.
> We propose multi-resolution inner cluster smoothing: this module effectively encodes long-range node relationships and keeps graph node number constant, enabling the model to leverage structural and latent adjacency matrices throughout the process. During our ablation study, this is shown to be much more effective than coarsening graph as in DiffPool, and can be potentially applied to other datasets beyond brain imagings.
> By applying integrated gradients on both input modalities (signal and structure), we interpret results and unveil patterns on both subject-, group-, and brain ROI-level. This can open up new opportunities for identifying biomarkers for different tasks or diseases.
>
> We also updated our submission with more supporting results in the revision. Thanks again for your time.

---

### Official Review · Reviewer_wRj8 · 2021-11-01

**Correctness:** 3
**Technical Novelty And Significance:** 3
**Empirical Novelty And Significance:** 3
**Recommendation:** 6
**Confidence:** 4

**Main Review:**

I found the approach introduced in the paper interesting, and the motivation is clear. I enjoyed that the proposed approach aims to extract latent dynamics information, thus capturing temporal heterogeneity, which can be very useful in neuroscience as well as disease prediction research. While this contribution shows that the proposed method can generate better prediction results, I see several issues with it as it stands. The rationale for combining various existing approaches as proposed in the paper should be clearly justified emphasizing the novelty. The presentation of the manuscript can be improved, and a few details seem to be missing. I have some concerns, which are listed below.

1) Existing approaches to estimate dynamic connectivity and approaches estimating spatiotemporal patterns in the fMRI data [1,2,3,4] are not mentioned. Authors could comment on what existing approaches (based on linear modeling) lack, relevance/irrelevance of the approaches in the given context, and compare the results.
2) The model is not straightforward to understand in one go. Paragraphs in Section 2.2 could be rewritten such that they are more connected and give a better picture of the model.
3) To validate the method and illustrate the motivation, please consider including some simulation studies on synthetic datasets. Since the method is intended to capture latent dynamics, the authors could borrow simulation settings from the related literature.
4) It is mentioned that the F1 metric is used because of the imbalanced dataset, but the information about the imbalanced dataset is missing for understanding the data. Also, the number of frames for each task is missing.
5) It seems that the experiments are performed only for one set of train, validation, and test set. Why are multiple runs of k-fold cross validation not used here?
6) Not sure what to make of the qualitative analysis in Figures 6, 12, and 13. It would be helpful to discuss the importance of "important" nodes in the context of these tasks rather than just stating the region name. Also, the authors should describe the regions mentioned in the main text. The claim of the paper is that the proposed method can find good representations but the authors have not explained the importance of these subnetworks in the context of the task presented. Even a small discussion would be helpful.
7) We notice that signal-important ROIs are not necessarily the same as connection-important ROIs:? What does it mean from the point of view of biological relevance? How to use these remarks as biomarkers?
8) How reproducible are these important regions extracted? Since the complete problem is non-convex, depending on initialization, will the end result vary? Since these regions would be used in further downstream analysis, they would need to be highly reproducible.
9) Authors mention, "We also find using adaptive adjacency matrices, and inner cluster smoothing can stabilize training, making the model less prone to overfitting and achieving close-to-best performance over a larger range of hyperparameters." Can you show results to support the claim made?
10) Authors mention, "Important brain regions obtained from $ATTR_A$ mostly comply with the previous literature." What are the important brain regions? Can you cite literature to back up this claim?
11) Is there a reason that $A_{iadp}$ is sparse for each sample? Is this expected biologically or due to the way in which the model is defined? Would it be helpful to use sparsity constraints on $A$? This might be helpful in controlling the sparsity and in the interpretability.
12) Can you comment on the biological relevance of the brain regions found in $ATTR_X$
13) How are $h_{adp}$ in eq 1 and $K$ in eq 2 set?
14) X-axis and Y-axis ticks size can be increased to improve the visibility in Fig 3, Fig 5, Fig 10 and Fig 11.
15) What is $\tilde{D}$ and $v$ in section 2.1? Is $\tilde{D}$ a diagonal matrix where $v,v$ entry is equal to the given sum?
16) Published versions of the referenced arxiv papers should be cited.

I am happy to raise my score if the above concerns are addressed.

[1] Li, Lingge, et al. "Modeling dynamic functional connectivity with latent factor Gaussian processes." Advances in neural information processing systems 32 (2019): 8263-8273.

[2] Taghia, Jalil, et al. "Bayesian switching factor analysis for estimating time-varying functional connectivity in fMRI." Neuroimage 155 (2017): 271-290.

[3] Zhang, Gemeng, et al. "Estimating dynamic functional brain connectivity with a sparse hidden Markov model." IEEE transactions on medical imaging 39.2 (2019): 488-498.

[4] Bhinge, Suchita, et al. "Extraction of time-varying spatiotemporal networks using parameter-tuned constrained IVA." IEEE transactions on medical imaging 38.7 (2019): 1715-1725.


**Summary Of The Paper:**

This paper proposes a Graph Neural Network model to estimate latent dynamics in the human brain using functional Magnetic Resonance Imaging (fMRI) and Diffusion Weighted Imaging (DWI) while performing a classification task. The model consists of four parts 1) estimating each sample's adjacency matrix by learning a shared projection matrix 2) gated temporal convolutional neural network for learning information from time-series data 3) graph neural network to get embeddings of temporal and spatial information combined by 4) multi-resolution inner cluster smoothing. The authors use integrated gradients for estimating important features for the classification task helping with model interpretation. The authors show that the proposed method can perform better in the prediction task than GCN, GAT V2, etc. On the real dataset, it is shown that the method can capture temporal and spatial heterogeneity and provide regions important for the presented classification task.

**Summary Of The Review:**

Based on the above assessments, I suggest rejection of this paper. The current experimental evaluations and discussion need to be stronger to be more convincing, and the presentation can be improved.

---

> ### Author Response · Authors · 2021-11-22
> **Response (part 1)**
>
> Thank you so much for your detailed comments, we really appreciate them. Here are the responses to your concerns:
> 1. Thank you for the suggestion, we limited our related literature to GNN-based fMRI modeling methods mainly due to the page constraint and found these are the most relevant ones. We briefly mention them in the revision.
> 2. We updated the section labeling and added the corresponding sections into the model figure. Texts are also slightly modified. Hopefully the model figure now can provide the needed connections between different parts of 2.2.
> 3. Thank you for the suggestion. We add simulation studies in A.2.7. We adapted another GNN library (https://github.com/alelab-upenn/graph-neural-networks) for graph time-series generation which is easier to control the ground truth to generate different classes. Detailed settings and results are added to the revision.
> 4. We added the number of scan sessions and frames per session for each task to Table2 in the appendix.
> 5. Thank you for raising the concern. The choice is made based on the number of settings in our ablation study (each setting requires grid search over multiple hyperparameter settings) and resource constraints. In Andrew Ng’s Stanford CS229 course, he mentioned that if sample number > 10k, one could use hold-out cross validation (plus an unseen test set). And this is our case when we generate inputs with sliding windows on the scanning sessions.
> To make the best out of this hold-out setting, we made sure training, validation, and test set each receive proportioned task sessions. For example, for task 1, with our 0.7-0.15-0.15 split, 0.7 * (number of task 1 samples) will be in the training set, and 0.15 * (number of task 1 samples) will be in the validation/ test set. The same is true for other tasks. This means test set won’t be filled with mainly one task. The only imbalance is different tasks have different numbers of samples, but this imbalance is the same for all training/validation/test sets.
> We also did k-fold cross validation with k=5 for length-256 inputs on our proposed model and ablation setting (vi) where no inner-cluster smoothing module is used. The relative F1 differences between the two settings for each fold are all roughly 10-11. So we believe our current results reflect the impacts of various settings/ methods.
> 6. Figure 13 (fig 14 in the revision) is to supplement Figure 5: showing the column sums of attribution of both real SC and random adjacency matrix in Figure 13 will give rise to similar patterns as in Figure 5. This is to argue that the model, especially with inner cluster smoothing modules, can learn the important signal sending regions relatively well even without explicit structures. However, to attribute real region connections in the brain, we need the attribution upon real SC (connections in the random adjacency matrix can be non-existent in the brains).
> And thanks for the suggestion regarding adding discussions about important regions! We add the result discussion with corresponding literature to the revision appendix section A2.5, and this should help with concern 10 as well.
> 7. Answered together with 12.
> 8. You are definitely right about the performance being slightly varied based on the initialization. We add a new section A2.6 (figure 17) in the appendix to show the attribution results are reproducible. In particular, we trained two models initialized randomly and on different data splits. Both models take in 256-frame inputs, and the validation weighted F1 for these two training partitions are 90.85 and 90.08. In addition, since the attribution method is meant to explain a trained model, it is beneficial to use the best-performing model for downstream analyses: explaining a good-performing model yields more reliable explanations than explaining a worse-performing one. That’s why we chose to use a model trained on longer inputs (they have better F1 than those trained on shorter inputs).
> 9. Thank you for the question, this was noticed both during training and during grid search for hyperparameters. We added the supporting results to Figure 11 under the appendix of revision.
> 10. We add the regions and related literature evidence to section A2.5 in the appendix.

---

> ### Author Response · Authors · 2021-11-22
> **Response (part 2)**
>
> (Continue from part 1 due to character limits)
>
> 11. Thank you for the question. Yes, after the model training, A_i_adp is naturally sparse for each sample. In particular, A_adp values range between (0, 1) due to softmax, and only around 2% of entries have values larger than 0.05. Even for the small values smaller than 0.05, most of them are extremely small and thus can be regarded as zeros (they cannot be really 0s because of the exp in softmax). See this (https://imgur.com/a/bqWuBSb) for the distribution of these small values (only showing values smaller than 0.1). For reference, the largest value can reach > 0.99.
> We think it is more model than biological induced. Similar A_i_adp sparsity patterns (most “non-zero” entries exist on a few columns of A_i_adp) exist for the synthetic data we tested on. Given how A_i_adp is used in GNN layers, each column represents a signal-originating ROI during message passing. We hypothesize that the model is learning the most effective “hubs” that pass information out to their neighbors. A related idea is information bottleneck (as in 'Deep Learning and the Information Bottleneck Principle' by Naftali Tishby): deep learning is essentially compressing the inputs as much as possible while retaining the mutual information between inputs and outputs. In a sense, A_i_adp represents the compressed hubs for a given input signal.
> As for the additional sparsity constraints, we tried adding L1 constraints on A_adp: Loss = CE + alpha * L1_norm(A_adp) / size(A_adp). However, this does not change the resulting performance in terms of F1 score nor A_adp sparsity. We tried with several alpha values as well. We hypothesized that the naturally trained A_adp is sparse enough, and further sparsification would move it away from the (local) optima thus won't be reached.
> 12. Based on how the model tasks the inputs, ATTR_A indicates important connections between brain regions, meaning the information passing between those regions is deemed critical for classifying task states. In contrast, ATTR_X indicates that the signals in those regions are important: it doesn’t matter if the signal activities leave the ROIs to connect with others, rather they themselves are useful in identifying the states.
> One observation from figure 17 of revision is that DYN and PVT have similar ATTR_A patterns, which means the (functional) connections originating from those regions (visual, control, and somatomotor systems) are important. But when looking at ATTR_X, they almost have some extreme opposites. For example, PVT has very high ATTR_X for ROI 15 (in LH_SomMotA), 30 (in DorsAttnA_TempOcc), 105 (in RH_VisCent_ExStr), etc. while DYN has very low ATTR_X for these ROIs. This suggests that the model is using activities on these regions to distinguish between different tasks: looking at signals on 15, 30, and 105 are helpful for identifying if they are PVT signals, but not so helpful for identifying them as DYN signals. To use our model to identify biomarkers, it is important to have “comparisons”, for example, different tasks, different disease states, etc. The attributed regions or connections are what the model is using for deciding if the inputs are in certain states. Therefore, the results are relative, instead of absolute, based on selected states to be compared against.
> 13. They are mentioned in the experiment section: h_adp is 5, and K is 2.
> 14. We updated the figures with larger tick font sizes.
> 15. Yes they are what you said. D~ is the diagonal node degree matrix, we modified the text to make it more clear.
> 16. Thanks for pointing this out. We were using google scholar generated bibtex, which seems not to work for some entries. We’ve double-checked all citations and cited the published version unless only the arxiv version exists.
>
> Again, thank you so much for these helpful comments and suggestions.

---

> > ### Comment · Reviewer_wRj8 · 2021-11-29
> > **Response to authors**
> >
> > Thank you a lot for writing the detailed rebuttal and for further improving the manuscript. I will increase my recommendation score to 5. While some comments were well addressed, a number of issues remained, such as
> > 1) Comparison with existing non-GNN based approaches since some work has already been done to estimate temporal patterns. This also relates to emphasizing the novelty of their work in relation to existing work.
> > 2) How to set $h_{adp}$ and $K$ experimentally?
> > 3) It is not always true that the best-performing model is the most reproducible and best explainable. If the results are used for downstream analysis, wouldn't a model be selected by balancing reproducibility and accuracy? Also, it appears that the regions are not highly reproducible; detailed comments on the reproducibility of the regions and the results are shown in section A2.6 would be helpful.

---

> > > ### Author Response · Authors · 2021-11-30
> > > **Thanks for the further comments**
> > >
> > > We thank the reviewer for increasing the score and for the further comments on our work.
> > >
> > > 1. We will add additional temporal signal classification baselines. The main difference will be using graph modeling and incorporating spatial relationships between regions can identify critical connections.
> > > 2. We tested h_adp = 2, 5, 10 in our experiments, and 5 appears to be the best so we chose this value for all the experiment settings. K = 1, 2, 3 were tested on a few settings, and K = 2, 3 have close performance, both outperforming K = 1. Given smaller K means smaller computation needs, we use K = 2 for all experiment settings.
> > > 3. Currently we do not have the data regarding the relationship between accuracy and reproducibility, but we will experiment to see if a tradeoff is needed. On the other hand, we think using more accurate models does lead to better explainability since the model is explaining correct predictions. We will verify this on synthetic data and add it to the simulation section. In addition, we found the attributions are in fact highly reproducible: in figure 17, only ATTR_X of DOT and MOD tasks have some deviations when derived from two models. Attributions from two models for other settings almost overlap with each other. Even for these two, important ROIs (those with high attribution values) are mostly consistent. Parcellation will also affect the result: when comparing subnetwork results based on the 17-network definition instead of 200-ROI parcellation, the attribution deviations become smaller. We will add comments regarding ROI (and subnetwork) attribution reproducibility to this section.

---

> > > > ### Comment · Reviewer_wRj8 · 2021-11-30
> > > > **Response to authors**
> > > >
> > > > Thank you for your response. I am raising my score to 6 based on further comments.

---

### Official Review · Reviewer_zFSB · 2021-11-03

**Correctness:** 3
**Technical Novelty And Significance:** 2
**Empirical Novelty And Significance:** 3
**Recommendation:** 6
**Confidence:** 4

**Main Review:**

Strength:
- Clearly written and easy to follow.
- Practically useful framework for fMRI data analysis.
- Extensive experiment that validates the proposed framework.

Weakness:
- Novelty of the paper is quite unclear, especially in the introduction. Why is analyzing time varying signals on graphs challenging? What are the problems that recent methods are facing, and how are the authors are solving them?
- Why is a normalized adjacency matrix used? Any technical / clinical motivation?
- It is not clear why having individual adjacency matrix implies a latent graph cannot be learnt. A_adp is a functional graph derived from node signals; why can't each adjacency matrix can be deployed here? This is important as the authors use both AHW and A_adpHW later in the framework.
- From anatomical perspective, relationships in local (structural) connections make more sense than those among far ROIs. What is the rational that it can be beneficial (rather than simply referencing Ying 2018)?
- In the experiment, did the authors divide the given dataset in to a single set of partitions? Was cross-validation adopted?
- Temporal data varies in their length per subject. How was this problem dealt?
- During the method comparisons, I am not sure if it is valid to set the same epoch limit to derive evaluation measures as each model has different complexity.
- Is the outcome properly validated? There are several labels according to memory, attention, dot prove and so on... are the label-specific variations detected from this framework really associated with those tasks?

**Summary Of The Paper:**

This paper proposes a deep learning method for temporal data on graph nodes, specifically designed for brain imaging data. It can be deployed for classification of data where the time-varying data and graphs are individual specific, and pinpoint subgraph structure where group-specific changes occur.

**Summary Of The Review:**

This paper tackles an important problem in neurosience.
Clinical motivation and the overall pipeline make sense, and extensive efforts were put on evaluation of the ideas and framework.
The paper is mostly clear, but I do have some concerns as mentioned in the Main Review above which can be very critical.
I am willing to change my score once I go through the rebuttal and other reviews if needed.

---

> ### Author Response · Authors · 2021-11-22
> **Response**
>
> Thank you so much for your time and comments. Here are the responses regarding your concerns:
> - Thank you for the question. Current deep learning neuroscience literature typically uses averaged fMRI when using graph neural network modelings, or structural information is not included. In contrast, we combine both imaging modalities (structural and dynamic functional signals) while modeling the temporal signals of fMRI efficiently. Our contributions are (1) We propose sample-level adaptive adjacency matrix learning based on input snippet, which can better capture changing input dynamics under different time or task states. (2) We propose inner-cluster smoothing: it effectively encodes long-range node relationships and keeps graph node number constant, enabling the model to leverage structural and latent adjacency matrices throughout the process, and (3) By applying integrated gradients on both input modalities (signal and structure), we interpret results and unveil patterns on both subject-, group-, and brain ROI-level. This can open up new opportunities for identifying biomarkers for different tasks or diseases.
> - Thanks for the question: the motivation is from GCN (Kipf17): if A is unnormalized, multiplication with A will change the scale of feature vectors thus exploding or vanishing values. Instead of row-normalizing with D^-1A, a symmetrically normalized A (D^-½ A D^-½ ) will not only take into account one’s own number of neighbors, but it’s neighbors’ number of neighbors. This gives more interesting dynamics as pointed out by Kipf and was used as a standard for later GCNs.
> - Sorry for the confusion, what we mean is: it can be troublesome to learn a universal latent graph A_adp for all samples like other standard approaches, because functional dynamics differ a lot from one input to another. A universal latent graph may not represent inputs well. Rather than that, we learn an individual latent adjacency matrix A_i_adp for each sample i. So the point is we can choose to learn a universal latent graph, but learning individual ones give rise to better representations of the data.
> - You are absolutely right that local connections should have more contributions from the anatomical perspective. We think the rationale behind adding soft cluster assignments being beneficial is that the local connections are captured well by the structural connectivity -- aka the input adjacency matrices. And upon graph convolution, local neighbors can pass their information to the neighbors effectively. Adding the cluster assignment and doing inner-cluster smoothing help complement the missing long-range relationships that are not expressed otherwise.
> - Thank you for raising the concern. The choice is made based on the number of settings in our ablation study (each setting requires grid search over multiple hyperparameter settings) and resource constraints. In Andrew Ng’s Stanford CS229 course, he mentioned that if sample number > 10k, one could use hold-out cross validation (plus an unseen test set). And this is our case when we generate inputs with sliding windows on the scanning sessions.
> To make the best out of this hold-out setting, we made sure training, validation, and test set each receive proportioned task sessions. For example, for task 1, with our 0.7-0.15-0.15 split, 0.7 * (number of task 1 samples) will be in the training set, and 0.15 * (number of task 1 samples) will be in the validation/ test set. The same is true for other tasks. This means test set won’t be filled with mainly one task. The only imbalance is different tasks have different numbers of samples, but this imbalance is the same for all training/validation/test sets.
> We also did k-fold cross validation with k=5 for length-256 inputs on our proposed model and ablation setting (vi) where no inner-cluster smoothing module is used. The relative F1 differences between the two settings for each fold are all roughly 10-11. So we believe our current results reflect the impacts of various settings/ methods.
> - Thanks for the question. As described in the experiment section, inputs are produced from sliding windows, with window lengths being 2^i (i = 3,...,8) for various experiment settings. So subjects generate different numbers of samples based on their differed length.
> - All methods are well converged within the 60 epochs (baselines typically converge around epoch 10) and we use the models with the best validation loss for test set inference.
> - Thank you for pointing this out, we compiled the results discussions to the revision appendix section A2.5.
>
> Hopefully these answer your questions and concerns about the paper.

---

> > ### Comment · Reviewer_zFSB · 2021-12-01
> > **Thanks for the comments.**
> >
> > While the authors answered most of my questions, I still have some concerns.
> >
> > - I don't see the motivation of this work properly described. Of course raising the performance of the model of a great interest, there is no motivation from the clinical perspective in the paper nor in the rebuttal. Why is it important to use fMRI and structural MRI together and why is it difficult to analyze time-varying signal on brain networks?
> > - Regarding normalized A, yes it makes sense from gradient point of view. However, if all the data are consistently and fairly obtained from the same scanner and the same protocol, is this normalization a necessary process? Is there any chance that the normalization will remove subject-specific characteristics?
> >
> > I do think that this paper definitely has technical contributions, but as a research complied from interdisciplinary area, I think the paper is quite biased on the technical side only.

---

> > > ### Author Response · Authors · 2021-12-01
> > > **Thanks for further comments**
> > >
> > > We thank the reviewer for reading our responses and revision and making further comments.
> > >
> > > Regarding the concerns:
> > > - Motivation: The main motivation is to generate a more complete representation of brain activities by combining multi-modality brain imaging data and the dynamical aspect of fMRI and efficient latent dynamics extraction. Higher model performance reflects better brain signal encoding ability, which can be beneficial to different downstream tasks such as disease prediction or traits prediction (e.g. if one has a good working memory).  In addition, in our experiments, we interpret the results with IG from various perspectives, and in the conclusion, we write “interpret the importance of both spatial brain regions and temporal keyframes (contributing to each task), as well as presenting heterogeneities among subnetworks, tasks, and individuals. These findings can potentially reveal new neural basis or biomarkers of tasks and brain disorders when combined with behavioral metrics, and enable more fine-grained temporal analysis around keyframes when combined with other imaging techniques”.  This is our more clinical side motivation, and we’ll stress it more in the introduction.
> > > - Using time-varying fMRI and structural MRI: We are using these data because our motivation is to generate a more complete brain activity representation: incorporating structural modality can provide extra connectivity information that is missing in the functional modality. The benefits of including it are also shown in the ablation study. As for the time-varying part, our argument is based on the limitation of current literature that analyzes fMRI signals with graph neural networks. They typically only use time-averaged fMRI or do not separately model spatial modality and the temporal one (in our experiments, we found in order to model time-varying brain signals well, the model needs to separate temporal modality and spatial modality instead of considering temporal signals as features attached on the spatial nodes). There exist works that model dynamic fMRI with Gaussian process, hidden Markov model, etc. but we approach the problem from a more data-drive perspective with a deep neural network and avoid using assumptions as in linear models.
> > > - Normalizing $A$: For a neural network to work with proper gradients, the $A$ needs to be normalized. This normalization in fact won't remove subject-specific characteristics because $A_i$ is normalized by its own diagonal degree matrix $D_i$ (as the equations written in section 2.1). Here $i$ is for $i^{th}$ sample only, so the normalization process won't rely on any other samples other than $i$. Therefore, subject-specific characteristics are not affected.
> > >
> > > Thank you again for your comments!

---

> > > > ### Comment · Reviewer_zFSB · 2021-12-01
> > > > **Thanks for the comments.**
> > > >
> > > > Thanks for the clarification.
> > > > I have raised my score, and I hope these comments get sufficiently reflected in the final manuscript.

---

### Official Review · Reviewer_Cuxq · 2021-11-03

**Correctness:** 3
**Technical Novelty And Significance:** 3
**Empirical Novelty And Significance:** 3
**Recommendation:** 5
**Confidence:** 4

**Main Review:**

Strengths:

- The proposed method combines structural and functional brain information, which are often assessed separately.

- The proposed network includes components for both temporal and spatial processing.

- The proposed network utilizes an individualized graph structure (connections) for each sample, rather than defining one adjacency matrix for all like most standard approaches. The network can handle this in part due to the soft clustering and keeping the same number of nodes at each GNN level.

- The paper is fairly clearly presented.

Weaknesses:

- The "strided non-causal TCN" I feel is a bit of a misleading name - when I think temporal convolutional network I imagine it needs to have the causal aspect, as it's trying to ensure appropriate analysis of temporal information. The "non-causal" TCN is just a strided convolution on 1D data that happens to be in time dimension. Also, it would be helpful if the authors briefly include the information/equation for how the gating mechanism works.

- For the experiments, the partition of the dataset appears to be by scan, not by subject. Since there are 1940 scans, 6 different scan tasks and only 56 subjects, my understanding is each subject may have undergone multiple session of the same task. Thus, it does not seem fair to split by scan, but rather should perform a patient-wise split for the validation method to get a better estimate of generalization performance.

- While the total number of scans is reasonably large for neuroscience study, given the smaller number of individual subjects, a single split validation study does not seem appropriate - a cross-validation framework would be more convincing. Furthermore, as there is only a single split, there is no variance information so it is difficult to assess if there are true differences between methods that perform similarly.

- The extremely large gap in empirical performance between the standard GNN models and the proposed method seems rather jarring (up to ~45% difference in accuracy). This makes me wonder whether the other models are not being appropriately tuned, or are given appropriate depth? I did not find any details about the baseline models, e.g., how many layers are used? For the proposed method it seems the authors would have used 4 layers, did they match the same number for baseline models? Also, how are the graphs constructed for the baseline model - are the same connections used or did the authors vary connections per sample like in their approach? And are the connections based on structural or functional data? In general, how would the authors explain the large difference in performance? It could also be interesting to compare to other GNN methods that were developed specifically for functional brain data analysis, e.g. Li et al (as cited in the paper), or Li et al, Braingnn: Interpretable brain graph neural network for fmri analysis, bioarxiv 2020.

- The authors compare to GNN-only baselines, but not TCN-only baseline. I think it would make sense to compare to TCN models given (over) half of their network is based on TCN, and also given the large gap in performance using GNN only models; this may point to the feature learning in the TCN layers as being very important compared to the GNN component for classification learning.

- How is the cluster number c chosen? The authors note that c decreases with deeper layers, but there is no mention as to the values used or how they are determined. Considering the authors claim this cluster smoothing is essential, it is important to explain how related parameter c is set. Does changing c then greatly affect the results?

Minor comments:
- For the variable for structural connectivity SC, the use of 2 letters for 1 variable is confusing when reading the equations - considering changing to 1 letter.
- More detail on the number of each type of task scan and how many scans per subject would be welcome (eg added to appendix).



**Summary Of The Paper:**

This paper presents a graph neural network architecture for analyzing dynamic brain imaging data. The proposed approach uses both structural connectivity (via DWI) and functional connectivity (via fMRI) to setup the graph edge information, while the fMRI signals are used to define node features. The proposed network alternates between analyzing temporal information via gated temporal convolutional network layers and spatial information via GNN layers. A multi-resolution soft cluster smoothing is applied as the pooling operation in the GNN. Attribution of structural and functional imaging inputs is also explored using integrated gradients interpretation method. The proposed method is tested on a multi-task fMRI dataset and demonstrated better performance in both ablation studies and comparison to state-of-the-art GNN methods.

**Summary Of The Review:**

My recommendation is based on the new proposed network for combining functional/structural brain data analysis in a meaningful way for processing temporal and spatial information, and allowing individualized graph structures to be learned/used within a GNN model. While the results appear good, they appear somehow incredulously higher than standard GNN methods, and important missing information about experimental setup and empirical comparisons further dampen my enthusiasm for the work.

---

> ### Author Response · Authors · 2021-11-22
> **Response (part 1)**
>
> Thank you so much for your time and comments. Here are the responses regarding your concerns:
> - Thank you for the question. TCN here means convolution along temporal dimensions. Our model is not autoregressive and signals are encoded into a single (temporal) point. We are trying to classify the input signals after seeing all of them which means y depends on x_t for all t. Typically, causal convolution is used for outputs of length T: y_0, … y_T, where y_t needs to only depend on x_0, … x_t. It’s somewhat similar to unmasked attention v.s. masked attention for the language modeling encoders.
> The gating mechanism is added into 2.2 (II), thanks for pointing it out.
> - Yes, you are right that the same subject will undergo multiple sessions (the session number of different subjects varies, ranging from 1 to 9). However, these sessions are happening across different days under different sleeping conditions. And upon examination, the signal distributions of the same subject across multiple sessions are different: the figure (https://imgur.com/a/YiO0vi4) shows ROI-0 (one of the VisCent regions)’s signal distribution of 2 sessions of subject 1 and 1 session from subject 2 during the visual working memory task. This is why we chose to split by scans.
> To double-check, we did an experiment that leaves out three subjects for testing and used the rest of the subjects to train (1769 training sessions and 171 testing sessions). When tested on the left-out subjects, the weighted F1 is 94.87. This shows our model is generalizable to scans of unseen subjects.
> - Thank you for raising the concern. The choice is made based on the number of settings in our ablation study (each setting requires grid search over multiple hyperparameter settings) and resource constraints. In Andrew Ng’s Stanford CS229 course, he mentioned that if sample number > 10k, one could use hold-out cross validation (plus an unseen test set). And this is our case when we generate inputs with sliding windows on the scanning sessions.
> To make the best out of this hold-out setting, we made sure training, validation, and test set each receive proportioned task sessions. For example, for task 1, with our 0.7-0.15-0.15 split, 0.7 * (number of task 1 samples) will be in the training set, and 0.15 * (number of task 1 samples) will be in the validation/ test set. The same is true for other tasks. This means test set won’t be filled with mainly one task. The only imbalance is different tasks have different numbers of samples, but this imbalance is the same for all training/validation/test sets.
> We also did k-fold cross validation with k=5 for length-256 inputs on our proposed model and ablation setting (vi) where no inner-cluster smoothing module is used. The relative F1 differences between the two settings for each fold are all roughly 10-11. So we believe our current results reflect the impacts of various settings/ methods.
> - Thanks for raising the concern. To search for baseline models, We went through all models in PyTorch Geometric (PyG) library and its Temporal (PyG-T) extension. They contain the most up-to-date and well-organized open-source graph neural network model implementations. For each selected baseline, the constructed model contains 4 corresponding graph layers for encoding, and the same decoder as ours (2 pooling layers on node axis, and 2 linear layers for classification), hyper-parameters are tuned with grid search. We added the structure detail to the revision.
> We are aware of the large performance gap. We think the most crucial reason is PyG treats temporal signals as feature vectors instead of putting them into a separated temporal dimension. Without sequence modeling on the temporal dimension, even the recently published graph attention models (GAT-v2 and graph Transformer) cannot give a good performance. As for PyG-T, almost all models in it assume one graph for the inputs (application scenarios are traffic network forecasting, link predictions, etc.), but we need to feed different adjacency matrices for every sample. Out of them, we were able to choose one model (GConvGRU) that supports different adjacency matrices, but it didn’t give a satisfactory result. Given this, we think our proposed sample-level adaptive adjacency matrix learning and multi-resolution inner cluster smoothing are useful in improving performance. This is also demonstrated with ablation studies.
> Lastly, we really appreciate your suggested GNN methods to compare with. We encountered the BrainGNN work during our baseline search but found they are not open-sourced. We will update the results if the implementation comes out and time permits.

---

> ### Author Response · Authors · 2021-11-22
> **Response (part 2)**
>
> (Continue from part 1 due to character limits)
> - Thank you for the suggestion! We did two experiments with TCN baselines and added them to the revision: one is replacing our current GCN part with CNN (1*1 kernels), the other is removing GNN altogether without any replacement. The test F1 on the former model (TNN+CNN) is 75.79, and the latter (TCN-only) is 71.98. Given that they outperform all other graph-related baselines, temporal modeling is definitely the more important part of the dynamic brain signal modeling process. However, involving spatial modeling with GNN still outperforms using CNN, which validates the structure of our model.
> - Thanks for the question. In our experiment, c values are halved for every inner-cluster smoothing module: the node number is 200, and c values are 100, 50, 25, … (exact number depends on how many GNN layers are used). This information is added to the revision. When using the smoothing module, changing c mostly affects convergence rate and only has minor effects on the final performance: we add corresponding information and figures to Appendix A.2.3 (III) in the revision.
>
> We also added the number of scan sessions and frames per session for each task to Table2 in the appendix. We really appreciate your comments and suggestions.

---

### Decision · Program_Chairs · 2022-01-20

**Decision:**

Reject

**Comment:**

This paper proposes a Graph Neural Network model to estimate latent dynamics in the human brain using functional Magnetic Resonance Imaging (fMRI) and Diffusion Weighted Imaging (DWI). The representation is tested on a classification task. While reviewers acknowledge the importance of this application, various concerns have been raised and partially addressed. The work focuses on graph deep learning and offers limited evidence of its superiority over more traditional ML or non graph based deep learning. Besides the methodological novelty is unclearly argued, which is not ideal for the audience of a conference like ICLR.

For all these reasons, this work cannot be endorsed for publication at ICLR 2022.